# Biophysical controls of marsh soil shear strength along an estuarine salinity gradient

Megan N. Gillen[1,2,3], Tyler C. Messerschmidt[4], Matthew L. Kirwan[4]

[1]Department of Geology, William & Mary, Williamsburg, VA 23187-8795, USA
5 [2]Earth, Atmospheric, and Planetary Science, Massachusetts Institute of Technology, Cambridge, MA 02139, USA
[3]Geology and Geophysics, Woods Hole Oceanographic Institution, Woods Hole, MA 02543, USA
[4]Virginia Institute of Marine Science, William & Mary, Gloucester Point, VA 23062-1346, USA

*Correspondence to*: Megan N. Gillen (mgillen@mit.edu)

**Abstract.** Sea-level rise, saltwater intrusion, and wave erosion threaten coastal marshes, but the influence of salinity on marsh
10 erodibility remains poorly understood. We measured the shear strength of marsh soils along a salinity and biodiversity gradient
in the York River estuary in Virginia to assess the direct and indirect impacts of salinity on potential marsh erodibility. We
found that soil shear strength was higher in monospecific salt marshes (5 - 36 kPa) than in biodiverse freshwater marshes (4 -
kPa), likely driven by differences in belowground biomass. However, we also found that shear strength at the marsh edge
was controlled by sediment characteristics, rather than vegetation or salinity, suggesting that inherent relationships may be
obscured in more dynamic environments. Our results indicate that York River freshwater marsh soils are weaker than salt
marsh soils, and suggest that salinization of these freshwater marshes may lead to simultaneous losses in biodiversity and
erodibility.

## 1 Introduction

Tidal marshes are rapidly evolving ecosystems that sit at the boundary between land and sea, and are influenced by a tight
coupling between biological and geomorphic processes (Redfield, 1972). Marshes provide a wide variety of ecosystem
services, such as improving water quality, sequestering carbon, and reducing the impacts of storm surge and coastal erosion
(Barbier et al., 2011). However, tidal marshes are vulnerable to climate change and its effects—such as sea-level rise, increased
storm frequency, and saltwater intrusion (Craft et al., 2009; FitzGerald et al., 2008; Kirwan and Megonigal, 2013).

Erosion of salt marsh edges is a primary cause of marsh loss (Fagherazzi et al., 2013; Ganju et al., 2017). Relative sea-level
rise potentially increases wave height, wave power, and edge erosion rates (Marani et al., 2011; Mariotti and Fagherazzi, 2010;
McLoughlin et al., 2015), but also leads to non-linear changes in vegetation that could alter the strength of eroding soils (Feagin
et al., 2009; Kirwan and Guntenspergen, 2012; Stagg et al., 2017; Wilson et al., 2012). Sea-level rise tends to increase lateral
erosion rates depending on wind direction and duration (Valentine and Mariotti, 2019). Numerical models of marshes have
shown that erosion rates and shoreline morphology depend on interactions between shear stress from waves and the shear

strength of marsh soils (Bernik et al., 2018; Leonardi and Fagherazzi, 2014; Marani et al., 2011; Mariotti and Fagherazzi, 2010).

While many studies have analyzed how sea-level rise and storms influence wave erosion (Howes et al., 2010; Leonardi et al., 2016), less is known about the processes driving the strength of marsh soils and their impact on erosion rates (Jafari et al., 2019). In general, vegetation increases the shear strength and decreases the erodibility of marshes (Ameen et al., 2017; Sasser et al., 2018; Wilson et al., 2012). Nevertheless, it is difficult to attribute large-scale controls on shear strength in marshes due to the heterogeneous distribution of roots, stems, soil types, and shells which all yield variable influences (Jafari et al., 2019). Previous work has examined shear strength in marsh soils in the context of vegetation, geomorphic setting, and methodology (Ameen et al., 2017; Jafari et al., 2019; Lin et al., 2016; Sasser et al., 2018; Watts et al., 2003; Wilson et al., 2012). However, additional research is required to fully understand the biophysical drivers of marsh soil shear strength.

Predicting how erosion rates will change in tidal marshes with increased salinization from sea-level rise requires a more mechanistic understanding of the drivers of shear strength. Salinity likely influences shear strength through ecological factors, such as dominant vegetation communities or distribution of belowground biomass (Ameen et al., 2017; Sasser et al., 2018). An increase in salinity may increase marsh soil shear strength assuming high salinity marshes favor species with deeper roots (Howes et al., 2010). Alternatively, shear strength may decrease with increasing salinity due to the loss of vegetation biodiversity (Ford et al., 2016). Here, we measure soil shear strength and biophysical parameters along an estuarine salinity gradient and use them to determine how salinity influences marsh soil shear strength.

## 2 Methods

### 2.1 Study area and approach

We measured the erodibility of marshes along a salinity gradient in the York River Estuary, a tributary of the Chesapeake Bay (Virginia, USA). The York River is a microtidal, partially-mixed estuary with a mean tidal range of 0.7 meters at the mouth of the river and 1 meter near the freshwater river sources (Fig. 1a) (Friedrichs, 2009; Sisson et al., 1997). The York River salinity gradient is created by saltwater from the Atlantic Ocean, and freshwater from the Mattaponi and the Pamunkey rivers (Reay, 2009). Sea-level rise rates are 3-4 times faster than the eustatic levels in this region, which could facilitate faster rates of salinization (Ezer and Corlett, 2012). Grain size on the river bed shifts from predominantly sand in the lower York River to a mud-sand mix in the middle and upper reaches of the estuary (Gillett and Schaffner, 2009). Various wetland types exist along the York River within different salinity regimes: polyhaline salt marshes that have monocultures of *Spartina alterniflora* (saltmarsh cordgrass), mesohaline brackish marshes with an extensive array of halophytic grasses, and oligohaline freshwater marshes with dominant plant species *Peltandra virginica* (arrow arum) and *Zizania aquatica* (wild rice) (Perry and Atkinson,

2009). Saltwater intrusion is driving an increase in salt-tolerant species at the freshwater marsh sites (Perry and Atkinson, 2009).

We chose five marshes along the York River salinity gradient for this study (Fig. 1a; Table 1): Goodwin Islands (salt), Catlett Islands (salt), Taskinas Creek (brackish), Sweet Hall Marsh (fresh), and the Pamunkey Indian Reservation (fresh). Salinity decreases upriver from 18 ppt at the Goodwin Islands to 0 ppt at the Pamunkey Indian Reservation (Reay, 2009). Within these overall sites, we chose sampling locations during a July 2018 survey cruise that followed the propagation of high tide along the York River. Sampling locations were selected based on similar flooding depths at high tide to maintain consistency in

inundation depths and along tidal creeks 5-10 m wide, with marsh widths beyond 20 meters. We collected samples from two zones within each marsh: (1) the tidal channel marsh edge located between the tidal channel and any levee (1 m from edge) and (2) the interior marsh locate at a measured distance of 10-15 m away from the edge site (Fig. 1b). All field work was done in July-August 2018, except for the collection of cores for belowground biomass at the Sweet Hall Marsh and Pamunkey Indian Reservation marsh edge sites (September 2018) and elevation profiles of the Pamunkey Indian Reservation (March

75    2020).

## 2.2 Measurements of shear strength, vegetation, and soil properties

We measured shear strength and a variety of biophysical characteristics at each location in this study. We used a Humboldt Shear Vane to determine the shear strength of marsh soils. Although the effectiveness of the shear vane in wetland soils is undetermined (Jafari et al., 2019), it remains the most widely used method to quantify wetland soil shear strength in coastal

science (Ameen et al., 2017; Howes et al., 2010; Valentine and Mariotti, 2019). We used a 1 m long shear vane, with a 10 cm-long head for relatively weak marsh soils, and a 5 cm-long head to measure relatively strong marsh soils. The shear vane was turned until the marsh soil was broken, and unitless values on the shear vane were converted to kPa using the manufacturer's conversion formulas (10 cm head: kPa=shear vane value*10*0.0625; 5 cm head: kPa=shear vane value*10*0.5). Each shear strength profile was one meter long with measurements taken at 10 cm intervals. Ten replicate profiles were taken within 1

meter of each other over a 10 meter distance parallel to the shoreline per marsh location at each study site, for a total of 100 total soil shear strength profiles along the York River.

We measured elevation profiles for all five sites using real-time kinematic (RTK) GPS across a transect from the marsh edge to the interior (Fig. 1b). For each site, five ground control points were taken on the marsh edge, 1 meter in from the edge, and

in the marsh interior to obtain more accurate elevations at our specific sampling locations. Edge elevations ranged between 0.1 – 0.5 m and 0.3 – 0.6 m in the marsh interior for all sites except for the Pamunkey Indian Reservation, which was lower in elevation than the other sites. This discrepancy is likely due to the other elevation profiles being measured during the growing season while the Pamunkey Indian Reservation profile was measured in early Spring. There is no vegetation cover in freshwater marshes along the York River during colder months (Perry and Atkinson, 2009), and the marsh platform may lose elevation

due to seasonal erosion or subsidence (Pasternack and Brush, 1998). The tidal range is also greatest at the Pamunkey Indian Reservation across the York River (Table 1) (Friedrichs, 2009), which may contribute to the loss of elevation capital during these barren seasonal periods.

Aboveground and belowground biomass were measured destructively. We collected standing vegetation from three replicate aboveground stem clip plots (25 cm x 25 cm) located within 1-2 meters of each other per marsh zone at each study site. Samples were counted for the total number of stems, dried to a constant weight, and separated by species to calculate species richness. Three replicate belowground biomass soil cores (15 cm diameter, 50-70 cm depth) were collected within the aboveground plots after destructive harvest at each location within sites (except for the edge site at the Pamunkey Indian Reservation where only 2 cores were taken), and used to measure belowground root and rhizome biomass. We sectioned cores into 10 cm increments and washed these segments over a 1 mm screen sieve. Live belowground biomass was separated based on color, turgidity, and buoyancy in water. Samples were then dried to a constant weight, and used to generate belowground biomass profiles for each study site.

Water content, dry bulk density, and organic matter content were determined from two Russian peat cores (2 cm diameter, 1 m depth) collected at each sampling location per study site. Cores were typically sectioned into 1 cm segments for the top 30 cm and at varying 2-5 cm intervals for the bottom 70 cm. Samples were dried to a constant weight, homogenized, and combusted for 6 h at 550°C to burn off organic material.

**2.3 Statistical analysis**

We conducted all statistical analysis in Microsoft Excel. Replicate measurements were averaged together to create composite profiles for shear strength and biomass data. We employed simple linear regression analysis to determine significant correlations between shear strength and biophysical drivers. $R^2$ and p-values were calculated for each relationship using the regression tool from the Microsoft Excel Analysis ToolPak. In linear regression analyses broken down by salinity type, we simply grouped together data points from study sites with the same salinity regime (Table 1). To test for significant spatial differences in shear strength, we used a two-way Analysis of Variance (ANOVA) with marsh type (i.e., salt, brackish, and fresh) and marsh zone as the primary treatments. Shear strength values were averaged at concurrent depths for (1) Goodwin Islands and Catlett Islands and (2) Sweet Hall Marsh and Pamunkey Indian Reservation to create composite profiles for salt and freshwater marshes, respectively, for the ANOVA.

# 3 Results

## 3.1 Shear strength

Shear strength measurements generally ranged from 0-36 kPa, but differed between locations within a site, and across sites. There was an observable trend of shear strength increasing with depth at freshwater marsh sites for the upper 30 cm of the soil profile (Fig. 2). For brackish and salt marsh sites, patterns of shear strength with depth were inconsistent. There was a large increase in shear strength below 30 cm at the Goodwin Islands edge location (Fig. 2a), which we attribute to measurements that were within the antecedent lithology (i.e. non-marsh soils). In analyses related to the effect of salinity on soil shear strength

(discussed below), we used depth-averaged shear strength values from the upper 30 cm of the soil profile. We selected the upper 30 cm as our window of averaging, because shear strength varied little with depth beyond 30 cm at most sites, it excludes any influence of antecedent parent material, and corresponds to typical vegetation rooting depths. This approach allows for comparisons between sites that are based on the same depth interval at each site.

Depth-averaged shear strength increased significantly with salinity across study sites for interior locations only ($R^2 = 0.81$; $p = 0.04$) (Fig. 3b). There was no significant relationship between shear strength and salinity for edge locations across all sites ($R^2 = 0.04$; $p = 0.74$) (Fig. 3a). At the high salinity sites (Goodwin Islands and Catlett Islands), shear strength values ranged from 5.3-36.0 kPa for the upper 30 cm of the marsh interior, with an overall interior average of 18.5 kPa. Interestingly, despite considerable differences in elevation (Fig. 1b), the shear strength values at the low salinity sites (Sweet Hall Marsh and the

Pamunkey Indian Reservation) were similar (Fig. 2), ranging from 3.6-8.0 kPa for the upper 30 cm of the marsh interior with an overall average of 5.4 kPa. These shear strength values were also substantially lower than those reported from the high salinity sites (Fig. 3b), indicating that the general trend discovered between shear strength and salinity in the marsh interior is unaffected by the elevation discrepancies.

While both marsh type ($p = 1.48e-9$) and marsh zone ($p = 5.62e-20$) yielded significant influence on shear strength values, the interaction between these variables was also significant ($p = 4.67e-11$). This result from the ANOVA indicates that the effect of marsh zone on shear strength varied with marsh type. Interior sites yielded higher values of shear strength than edge sites in the brackish and the salt marshes (salt: 5.1 kPa at edge, 18.5 kPa in interior; brackish: 4.5 kPa at edge, 16.6 kPa in interior) (Fig. 4). There was a negligible difference between edge and interior shear strength values at the freshwater marsh sites (5.41

kPa at edge, 5.38 kPa in interior) (Fig. 4).

## 3.2 Biophysical drivers

In the marsh interior, vegetation properties largely explained variability in soil shear strength. Belowground biomass had the most significant influence on shear strength in the interior for salt and brackish marshes ($R^2 = 0.58$, $p = 1.09e-5$) (Fig. 5). Aboveground biomass (data not shown) was also correlated with shear strength in the marsh interior but was marginally

significant ($R^2 = 0.64; p = 0.105$). At the marsh edge, soil properties explained most of the variability in shear strength on edge sites. Water content was significantly correlated with edge shear strength values ($R^2 = 0.76, p = 5.72e-14$) (Fig. 6a). However, other properties that co-varied with water content were also important, including the relationship between organic content and shear strength at edge sites in salt marshes (Fig. 6b).

## 4 Discussion

The results from this study are consistent with previous work that identifies vegetation and soil properties as important drivers of marsh soil shear strength. For example, soil shear strength is well known to vary with dominant plant species (Howes et al., 2010; Sasser et al., 2018). Our work confirms this concept and finds that soil shear strength is positively correlated with belowground biomass in the marsh interior (Fig. 5). This finding aligns with natural and manipulative experiments that show shear strength increases with belowground biomass (Wilson et al., 2012), and that the mortality of belowground roots and

rhizomes is related to enhanced erosion (Coleman and Kirwan, 2019; Lin et al., 2016; Silliman et al., 2019; Wilson et al., 2012). Like previous work (Ameen et al., 2017; Wilson et al., 2012), our results demonstrate that soil properties such as water content and organic content are also important drivers of potential marsh erodibility (Fig. 6). However, we uniquely show that the relative importance of vegetation and soil properties depends on the location within a marsh (edge vs. interior).

The relationship between belowground biomass and shear strength was not significant at our freshwater marsh sites (Fig. 5b). However, this is likely due to the overall lower amount of belowground biomass present in York River freshwater marshes that would not produce a significant linear relationship compared to the range of biomass values found in our salt and brackish sites. Therefore, we maintain that differences between belowground biomass drive shear strength values in the marsh interior regardless of salinity, where low biomass values relate to low shear strength values both within a soil profile and across

different marsh types.

    Our work indicates that the marsh interior has a higher soil strength than the marsh edge at our salt and brackish marsh sites (Fig. 4). We ascribe this variability in saline marshes to biological drivers influencing marsh interior soils (Fig. 5), and soil properties determining soil shear strength at the seaward marsh boundary (Fig. 6). The differing influences are due primarily

to various processes occurring at different places in the marsh. Sedimentation is low in the marsh interior leading to more compacted stronger soils, and biomass tends to be concentrated closer together with higher stem densities. The tightly packed belowground root network adds cohesion to marsh soils without the active edge processes frequently reworking sediment (Silliman et al., 2019). In contrast, the marsh edge is typically more dynamic, where increased inorganic sediment deposition and resuspension leads to more unconsolidated, mineral-rich soils that can impact soil cohesion and shear strength (Ameen et

al., 2017). Low concentrations of belowground biomass present at the marsh edge (Fig. 5a) in tandem with processes actively reworking sediment may also contribute to lower soil shear strength values (Silliman et al., 2019). Enhanced nutrient loading,

particularly in wetlands undergoing eutrophication, at the marsh edge weakens soils and may also influence shear strength variability at the seaward boundary (Johnson et al., 2016; Turner et al., 2020; Wigand et al., 2018). It is unclear why similar spatial patterns were not observed in the freshwater marsh locations. Perhaps the overall lower belowground biomass and shear strength of the freshwater marshes precludes our ability to detect patterns across the marsh. Nevertheless, our findings indicate that belowground biomass drives soil shear strength variability in the marsh interior (Fig. 5), and soil properties influence marsh edge shear strength (Fig. 6).

Salinity potentially plays an important role in determining the erodibility of marsh soils, through its combined influence on vegetation type and belowground biomass. Prior work examining the relationship between salinity and marsh soil strength concludes that salt marshes are more resistant to lateral edge erosion than freshwater marshes, ascribing variation in rooting depth to differences in soil shear strength (Howes et al., 2010). While our study also finds that salt marshes are stronger than freshwater marshes in the marsh interior (Fig. 3), we find that salt marshes are generally stronger than freshwater marshes regardless of the depth within the soil and rooting profile (Fig. 2b). This relationship between salinity and shear strength may be also species dependent—while our *Peltandra virginica*-dominated freshwater marsh sites had the weakest soils, previous work shows other freshwater grass species such as *Panicum hemitomon* with relatively high shear strength values (Sasser et al., 2018). We therefore attribute stronger salt marsh soils in our study area simply to the greater belowground biomass of *Spartina alterniflora* relative to the freshwater species present along the York River, such as *Peltandra virginica* and *Zizania aquatica*.

Root structure and geometry may also have considerable influence over marsh soil shear strength (Ameen et al., 2017; Howes et al., 2010). In the salt and brackish marshes, *Spartina*-dominated systems, the belowground root network consists of fibrous, tightly interlocking strands that may hold soil more effectively (Fig. 7a). *Peltandra virginica* is the most abundant plant species in the freshwater sites examined in this study, whose belowground biomass consists of large tubers with aerenchyma tissue and easily broken roots spaced out throughout the marsh (Fig. 7b). This laterally heterogeneous distribution of roots across the surface of the freshwater marsh may lead to overall decreased soil shear strength. Investigations into root structure and geometry may also clarify how species type influences the relationship between belowground biomass and marsh soil shear strength (Sasser et al., 2018), and should be incorporated into future work.

Interestingly, we found no relationship between salinity and soil shear strength at the marsh edge, where erosion would actually occur (Fig. 3a). Although this finding warrants more attention, we suggest that processes (e.g., sediment deposition, erosion, and resuspension) and environmental conditions (e.g., low belowground biomass, eutrophication, etc.) associated with a more dynamic marsh edge obscure patterns that would otherwise be evident. Furthermore, other ecogeomorphic interactions between biophysical parameters unexplored in this study may have considerable influence on marsh soil shear strength. For example, marsh elevation and its effect on hydroperiod could have influenced biomass and soil properties within and across our sites

(Kirwan and Guntenspergen, 2012; Morris et al., 2002). Marsh elevation is in turn controlled by a number of processes such as organic and mineral accretion, compaction, and erosion (Baustian et al., 2012; Kirwan and Megonigal, 2013; Morris et al., 2002). Thus, it is difficult to understand the effect of each biophysical parameter in isolation, and the primary direction of influence. Future work should consider the interplay between dominant plant species, belowground biomass and root structure,
soil type, and hydrogeomorphic setting, their effect on each other, and on marsh soil shear strength.

Previous work in U.K. marshes finds that freshwater marshes with a more diverse array of vegetation species have stronger marsh soils due to greater belowground biomass (Ford et al., 2016). However, our study finds that marsh soil shear strength increases with a decrease in plant biodiversity. Like other estuaries (Brock et al., 2005; Engels and Jensen, 2010; Grenier La
Peyre et al., 2001; Odum, 1998), biodiversity decreases from freshwater sites to salt marsh sites in our study area (species richness = 6 for Pamunkey Indian Reservation, species richness = 1 at Goodwin Islands and Catlett Islands). Salt marshes in the York River are dominated by *S. alterniflora* (Table 1), which is a highly productive species that creates a dense network of belowground biomass (Perry and Atkinson, 2009; Silliman et al., 2019). We suggest that the overwhelming influence of this highly productive salt marsh species explains the high shear strength of less biodiverse marshes, as this species is absent
in our freshwater York River estuary sites and those studied in the U.K. (dominant species: *Puccinellia maritima* and *Juncus gerardii/maritimus*). While there was a positive correlation between shear strength and belowground biomass in both the York River and Essex/Morecambe Bay marshes, the relationship between biodiversity and belowground biomass differs. We find that a decrease in biodiversity leads to an increase in belowground biomass, while the previous work in the U.K. determines the opposite trend (Ford et al., 2016). While biodiversity may not be driving soil shear strength variability in this study, it may
be a more critical factor in the typically low productivity, high salt marsh platforms present across the U.K. (Ford et al., 2016). This difference in plant community between the York River and Essex/Morecambe Bay marshes additionally underscores the potential importance of species composition in driving variation in shear strength (Sasser et al., 2018), and should be investigated in tandem with biodiversity in future studies.

Sea-level rise and saltwater intrusion are impacting wetlands in the York River estuary (Perry and Atkinson, 2009), and globally (Herbert et al., 2015; Neubauer, 2013; Noe and Zedler, 2000). While accelerated rates of sea-level rise could enhance wave erosion (Mariotti and Fagherazzi, 2010), increase inundation of marshes, and threaten their survival (Kirwan and Megonigal, 2013), sea-level rise also leads to changes in vegetation type and productivity (Donnelly and Bertness, 2001; Kirwan et al., 2009; Morris et al., 2002). Although these changes will have a variety of ecological and geomorphic
consequences, our work suggests that saltwater intrusion alone could be accompanied by stronger salt marsh soils that are less easily eroded.

**Data Availability**

Data has been archived to the Long Term Ecological Research repository, and can be found at
doi:10.6073/pasta/26c848ab288cc14a2edb106f5800cfc8.

**Author Contributions**

All authors contributed to the organization of the study. MNG and TCM conducted field surveys and sample processing in the lab. MNG performed data analysis. MNG composed the manuscript with significant contributions from MLK. All authors
edited the paper.

**Competing Interests**

The authors declare they have no conflict of interest.

**Acknowledgements**

The authors would like to thank the two anonymous referees for their useful comments that helped improve this manuscript. We would also like to thank the Pamunkey Indian Tribe and the Chesapeake Bay National Estuarine Research Reserve System for access to field sites. The authors are grateful to Dan Coleman for conversations that improved this work, and to Alex Smith, Rosemary Walker, and Alexis Jenkins who helped with field and lab work. This research was supported by the U.S. National
Science Foundation awards 1654374, 1426981, 1529245, and 1832221. This is contribution no. XXXX of the Virginia Institute of Marine Science.

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

**Figures**

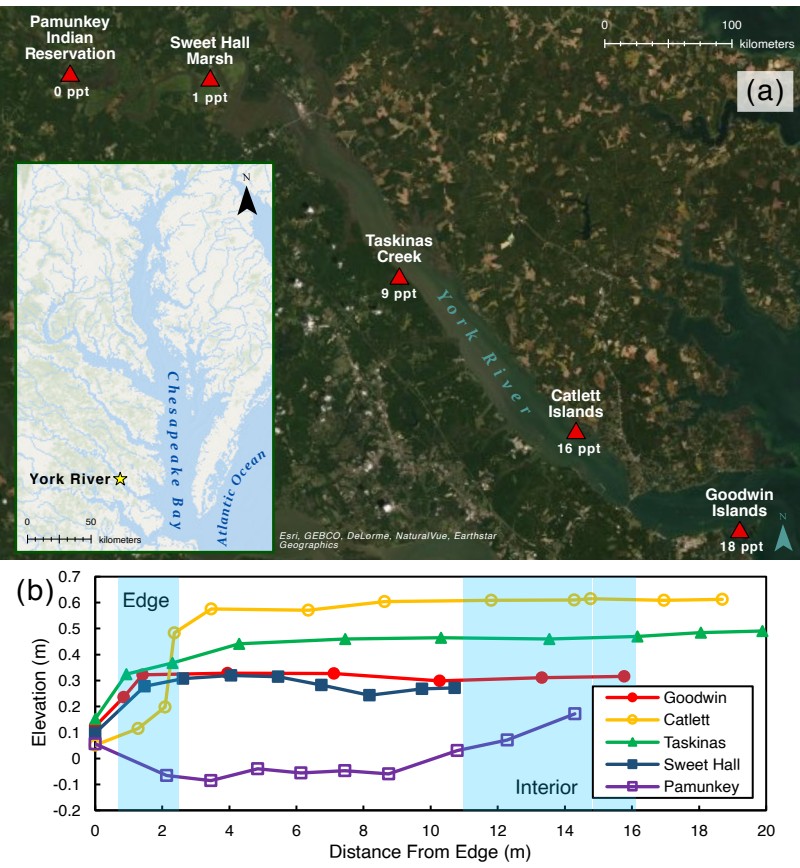

**Figure 1:** (a) Map of the York River Estuary study area, highlighting specific study site locations and their salinities. Average channel salinities are from Reay, 2009. (b) Elevation profile for each study site where elevation is relative to NAVD. Blue shading indicates the location of marsh edge and interior sampling locations.

| Study Site | Marsh Type | Tidal Range | Species Type | |
|---|---|---|---|---|
| | | | Edge | Interior |
| Goodwin Islands | Saline (18 ppt) | 0.7 m | *Spartina alterniflora* (tall form) | *Spartina alterniflora* (short form) |
| Catlett Islands | Saline (16 ppt) | 0.75 m | *Spartina alterniflora* (tall form) | *Spartina alterniflora* (short form) |
| Taskinas Creek | Brackish (9 ppt) | 0.85 m | *Spartina alterniflora* (tall form) | *Spartina alterniflora* (short form), *Spartina patens*, *Distichlis spicata* |
| Sweet Hall Marsh | Fresh (1 ppt) | 0.75 m | *Peltandra virginica, Zizania aquatica, Polygonum* species | *Peltandra virginica, Zizania aquatica, Spartina cynosarouides* |
| Pamunkey Indian Reservation | Fresh (0 ppt) | 1 m | *Peltandra virginica, Zizania aquatica, Scirpus* species, *Polygonum* species | *Peltandra virginica, Zizania aquatica, Polygonum* species, *Bidens laevis, Scirpus* species |

**Table 1.** Ecological and physical characteristics of the five selected sites for this study. Salinity data is average channel salinities from Reay, 2009. Tidal range data is from Friedrichs et al., 2009 and Sisson et al., 1997. Species data is from aboveground biomass samples collected for this study.

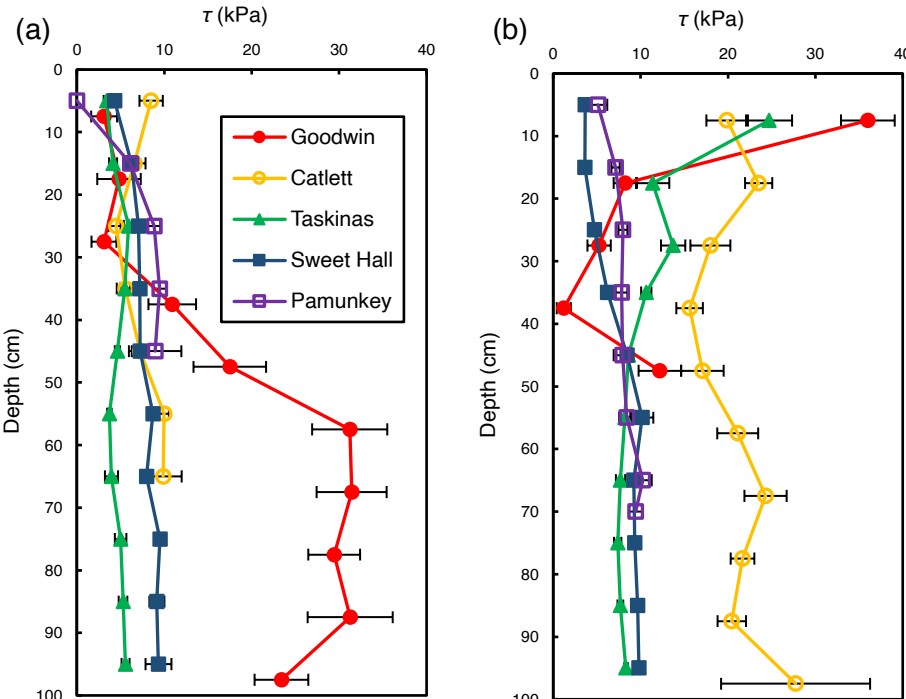

**Figure 2:** Average shear strength ($\tau$) profiles for marsh edge (a) and marsh interior (b) locations at each study site. Each profile represents the average of 10 replicate profiles, where error bars represent standard error.

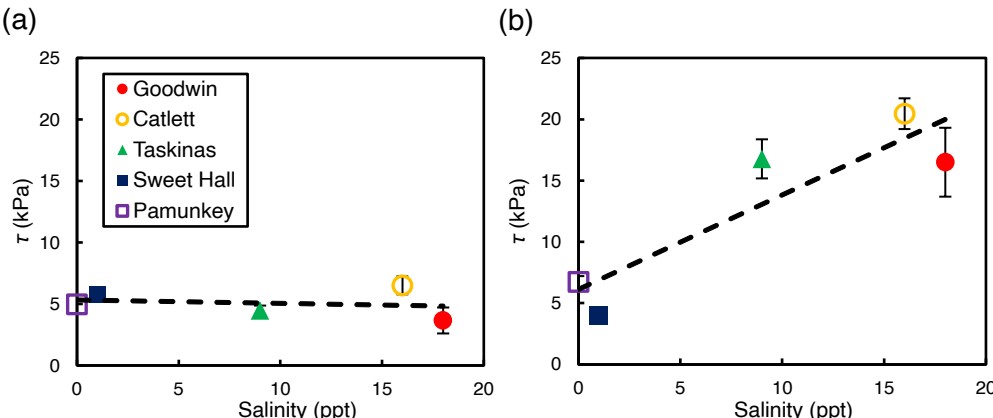

**Figure 3:** Relationship between salinity and shear strength ($\tau$) for edge sites (a) and interior sites (b). Each point represents the depth-averaged shear strength value for the upper 30 cm of the soil profile, where error bars represent the standard error associated with 10 replicate profiles for each site and location. Only the interior sites yielded a significant relationship between shear strength and salinity ($R^2 = 0.81; p = 0.04$).

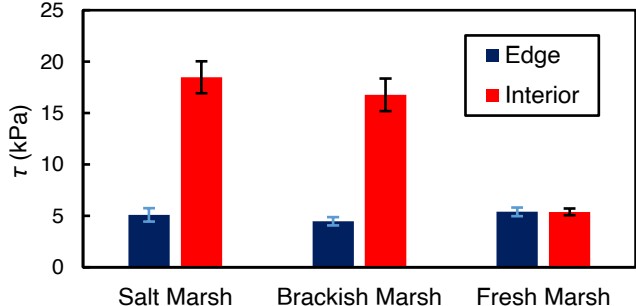

**Figure 4:** Average shear strength ($\tau$) values for marsh edge and interior locations. Salt marsh locations are Goodwin Islands and Catlett Islands, brackish marsh location is Taskinas Creek, and freshwater marsh locations are Sweet Hall Marsh and the Pamunkey Indian Reservation. Each column represents the depth-averaged shear strength for the upper 30 cm of soil of each marsh type. Error bars represent the standard error of 10 replicate shear strength profiles for each marsh type.

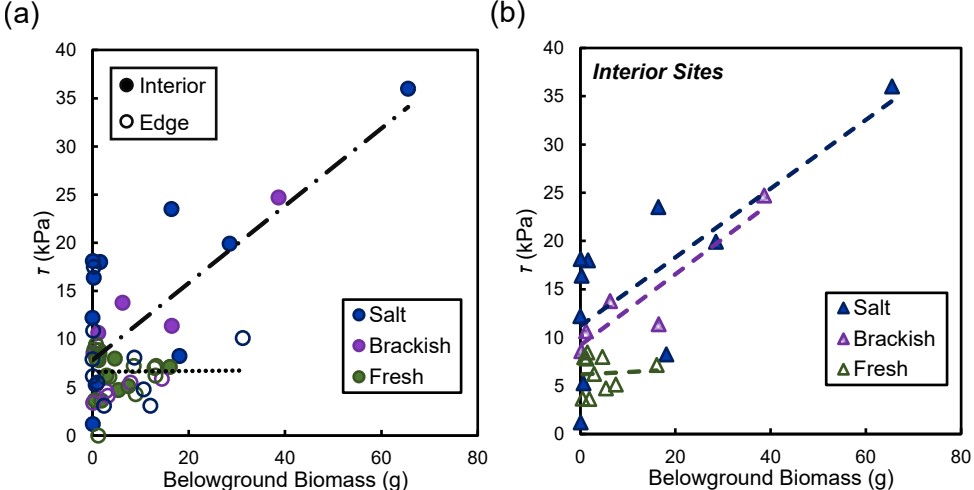

**Figure 5:** (a) Shear strength ($\tau$) compared to live belowground biomass for edge (open circles) and interior (closed circles) locations, broken down by salinity type (salt = Goodwin and Catlett, brackish = Taskinas, fresh = Sweet Hall and Pamunkey). Each point represents a biomass measurement with its associated shear strength value at concurrent depths in the soil profile. Only the relationship between belowground biomass and shear strength in the interior was significant ($R^2 = 0.58; p = 1.09e$-$5$). (b) Relationships between belowground biomass and shear strength ($\tau$) from (a) for interior sites only, broken down by salinity type (same as (a)). Only the relationships between salt ($R^2 = 0.57; p = 0.012$) and brackish ($R^2 = 0.86; p = 0.025$) sites were significant.

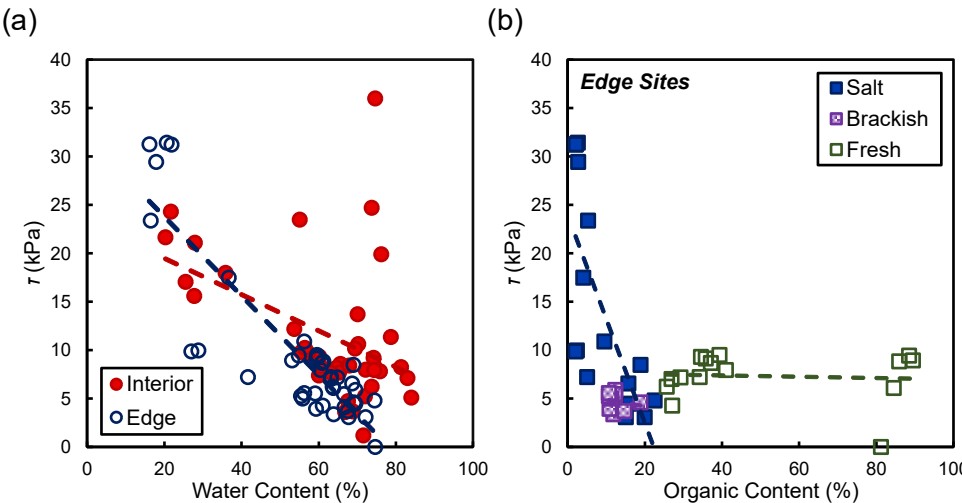

425

**Figure 6:** (a) Shear strength ($\tau$) compared to water content at concurrent depths in the soil profile. Only the relationship between water content and shear strength at the marsh edge was significant ($R^2 = 0.76$, $p = 5.72e\text{-}14$). (b) Shear strength and organic content at concurrent depths in the soil profile grouped by salinity type. The relationships between shear strength and organic content are as follows: salt marshes (Goodwin Islands and Catlett Islands) ($R^2 = 0.52$; $p = 0.001$), brackish marsh

430 (Taskinas Creek) ($R^2 = 0.04$; $p = 0.596$), and freshwater marsh (Sweet Hall Marsh and the Pamunkey Indian Reservation) ($R^2 = 0.01$; $p = 0.800$).

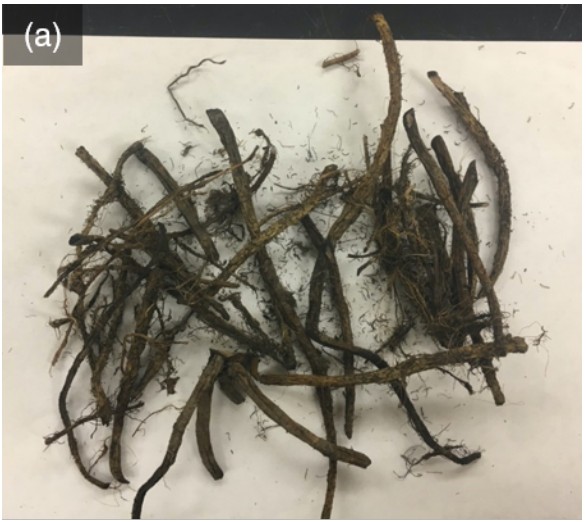

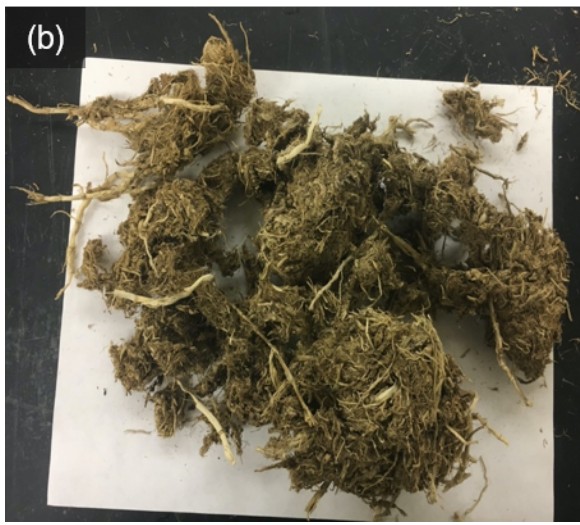

**Figure 7:** Photographs of belowground biomass root networks at concurrent depths for freshwater species *Peltandra virginica* (a) and salt and brackish marsh species *Spartina alterniflora* (b).