# Peer review of "Biophysical controls of marsh soil shear strength along an estuarine salinity gradient"

_Earth Surface Dynamics, 2020_

## Referee Comment (RC1) · Anonymous Referee #1 · 29 Sep 2020

This paper investigates the influences of salinity and belowground biomass on soil shear strength in coastal wetlands of the Chesapeake Bay area. It is well-organized and clearly written. The data used and methods of analysis are, in themselves, sound. However, I would like to suggest a few changes that would improve its clarity, and also point out what I consider to be weaknesses in their overall conclusion that, broadly speaking, tidal freshwater wetland soils have reduced soil shear strengths compared to their salt marsh counterparts.

1) I may have missed it earlier in the document, but I could not find in the text how the five sites were classified as either "fresh", "brackish", or "saline" until the caption for Figure 6. Prior to that figure, numerous references are made regarding these marsh types, but it is difficult to follow along with the data presented without knowing which

sites were classified as which marsh types. Please specify this classification earlier in the paper (ideally in the methods/study site section).

2) The bulk density values reported often seem quite high (many exceed the density of granite, commonly taken as 2.65 g/cm3). Are the bulk density values reported in the manuscript "wet" or "dry"? If they include water weight ("wet"), that would explain the high values.

3) The elevations measured at the Pamunkey site were much lower than those measured at the other sites. The authors explain this difference as possibly arising from the Pamunkey elevations being surveyed early spring prior to vegetation growth, while elevations at the other sites were measured well into the growing season. I suppose that makes sense to some degree, but the difference approaches 50 cm (through examination of Figure 1). That seems like a lot of elevation capital to attribute to a seasonal biomass cycle, and whether or not this site is considerably lower than the other site has fairly strong implications for interpretation of some of the differences seen in the data. It would be helpful if the authors could explain how before/during growing season differences could explain this much elevation difference in the marsh soil surface, which is presumably what was surveyed, and specifically what processes are at play here.

4) Following up on comment (3) above, numerous studies exist that illustrate the importance of inundation in controlling belowground biomass. Many show hump-shaped responses, where high sites and low sites exhibit reduced biomass/productivity, and optimum production occurs at intermediate elevations. Others show monotonic production/biomass decreases with decreased elevation (or increased flooding). There is no mention of those dynamics to speak of in this manuscript. The authors should interpret their findings to some degree in the context of those studies. This is particularly the case given that their sites consistently show higher elevations at the "interior" locations than at the "edge" locations. Those differences are not trivial – mean interior elevations exceed their edge counter parts by 11cm, 14cm, 15cm, 11cm and 20cm for Pamunkey, Sweet Hall, Taskinas, Catlett, and Goodwin locations, respectively. I'm unfamiliar with

how those elevation differences would translate into differences in hydroperiod in the Chesapeake Bay region, but in the microtidal, low gradient wetlands on the northern Gulf of Mexico coast, those elevation differences would easily translate into differences in flood duration that would exceed 30 % (e.g., an interior marsh at 0.25 m NAVD that was flooded at 40% of the time would be flooded around 70% or more of the time if its elevation were reduced by 0.15 m).

5) In Lines 151-153, enhanced nutrient loading at the edge sites relative to the interior sites is suggested as a possible explanation for why shear strength at the edge sites may be lower. However, according to figure 1, this distance is only on the order of 10 m. It does not seem reasonable to expect a meaningful reduction in nutrient concentrations over this length scale, but perhaps the authors can demonstrate otherwise with citations to support this claim.

6) The authors cite the Howes et al. (2010) paper that concludes that "salt marshes are more resistant to later edge erosion than freshwater marshes" (Lines 158-160, this manuscript). The Howes et al. (2010) paper identified Spartina patens as the dominant vegetation present in the low shear strength region of Breton Sound basin, Louisiana, that was so badly decimated by the shearing forces of Hurricane Katrina. They pointed out that although this species "has extensive rooting but of smaller diameter [than S. alterniflora]. The plant is less tolerant to anoxic soil conditions, which likely limits the rood network to shallower depths" and use this logic to conclude explain why S. patens regions were sheared while S. alterniflora regions were largely intact after the storm. However, in lines 180-185 of this manuscript, the authors cite the presence of S. patens (and the co-occurring S. alterniflora) as the reason the salt marshes of the York River estuary exhibit the relatively high shear stresses, owing to their high productivity and their creation of dense networks of belowground biomass. It is difficult to square these – Howes et al. (2010) on the one hand indicating that it is the limited development of the S. patens root network that contributes to a low shear stress, and this manuscript on the other hand citing the dense root network of S. patens in promoting high shear

strength. It is important to consider, when citing Howes et al. (2010), that their "low salinity", "low shear strength" zones were vegetated almost exclusively by S. patens. In the present study, S. patens is identified as being in the high shear, high salinity zone.

7) Given the rather large elevation gradients across each of the transects (see comment 3 above), it seems reasonable that vegetation species composition varies markedly across the transects (particularly between the "edge" and "interior" zones. Given that a paper cited frequently in this manuscript (Howes et al. 2010) attribute variations in shear strength to taxa-specific morphological differences in root structure, if transitions in species composition existed across the elevation gradients in the transects, within-transect species composition could be responsible for some of the patterns observed in the manuscript. Can you speak differences in species composition not only between the transects, but within each transect as well?

8) Similar to what was done for figure 6, could the regression in Figure 5 be separated by marsh type? For example, when breaking this regression down by site (see attached figure), a different picture emerges – biomass vs. shear strength is significant for Taskinas (the brackish site) and Goodwin (one of the saline sites), but insignificant for the remaining three sites. Beyond simple reductions in sample sizes and degrees of freedom, are there other explanations for why this relation may be significant at some of the sites (or marsh types) but insignificant at other sites?
* * *
**Dashed Line indicates Regression not Significant**

Figure: BGB vs Shear Strength scatter plot with regression lines for Pamunkey, Sweet Hall, Taskinas, Catlett, and Goodwin sites. X-axis: BGB (0–70). Y-axis: Shear Strength (0–40).

Legend:
- Pamunkey (purple dashed)
- Sweet Hall (blue dashed)
- Taskinas (green solid)
- Catlett (yellow dashed)
- Goodwin (red solid)

**Fig. 1.** Figure 5, BGB vs Shear Strength, by site

---

## Referee Comment (RC2) · Anonymous Referee #2 · 23 Nov 2020

General Comments: This reviewer believes that the data presented in the manuscript is of value and should be publishable. However, there are numerous problems (see below) that first need to be addressed.

Specific Comments: 1. Line 11: "erodibility" was not directly measured in this study. Insert "potential" between before "marsh erodibility". 2. Line 12: Insert "likely" before driven. 3. Line 13: "rooting structure" was not measured in this study. Hence, the need for comment 4. Lines 15-16: The last sentence in the abstract is misleading and not supported by the data presented in the manuscript. Based on the data presented in the manuscript, one can only state that the freshwater marshes investigated had weaker soils than the salt marshes investigated. Extrapolating to salt marshes and fresh marshes universally goes far beyond this specific study, and, based on Table 3

[Figure]

in Sasser et al. (2018), is an unsupported generalization. 5. Line 65: the statement is made that there "were consistent elevations across all study sites" and Fig. 1b is referenced. However, when this reviewer looks at Fig. 1b, one sees elevations that differ both among the marshes and within a marsh. Hence, this statement appears incorrect and may need qualification. 6. Lines 65-68: The reviewer suggests rewriting these lines as follows: We collected samples from two zones within each marsh: (1) the tidal channel marsh edge located between the tidal channel and any levee (1 m from edge) and (2) the interior marsh located at a measured distance of 10-12 m away from the edge site (Fig. 1b). 7. Lines 78-79: Provide information on the spatial distribution of the 10 replicated profiles. Were they within a 1-meter radius, for example, or where they distributed over a 10 m distance parallel to the shoreline, for example? 8. Lines 83-84: It is stated that "All sites were located at similar elevations …….." . , However, Fig. 1b shows that, for example, the Catlett study site had dissimilar elevations within the edge zone. Maybe what would help is to provide the range in elevations for the specific zones in each marsh where shear strength was measured. A table might also work. The bottom line is that the authors do not adequately convince this reader that elevations were "similar". 9. For both above- and belowground sampling, the spatial distribution of the three replicates should be provided. Were they within a meter of each other or 10 meters of each other? 10. Generally, soil shear strength is very much dependent on the plant species that occur where measurements are taken. Knowing the dominant community type does not provide enough information. Hence, within the Methods or Results, the authors should specify the specific plant species present where the shear strength vanes were inserted. In the salt marsh, of course, it is likely to be Spartina alterniflora, but in the brackish marsh, it could be any number of species (e.g., S. patens, S. cynosuroides, or Distichlis, etc.), and in the fresher marshes it could be a mixture of species or one dominant like Peltandra. This information is needed to better interpret the shear strength results. 11. Lines 57-101: This reviewer is surprised that soil textural components were not measured. Are sand, silt, and clay data available for these study sites? 12. A Statistical Analysis sub-section is

absent from the Methods section. The authors should describe the experimental design of their study and the statistical methods used to test significance. In this regard, this reviewer finds it difficult to understand why the authors did not use an Analysis of Variance approach to identify significant differences among study sites and between zones, as well as their interaction, which they graphically present in Figure 4 and discuss in lines 119-123. See below for more on this. 13. Lines 119-123: Figure 4, which is discussed in lines 119-123, may be the most important results of the study. Yet, it received no statistical analysis. Although one might argue that the study is pseudoreplicated, given that the York River salinity gradient was not replicated i.e., two or more rivers), one can still do a 2-way factorial ANOVA with study site (the five marshes) and marsh zone (edge and interior) as the main treatments. The marsh zones should be nested within study sites. The authors could even include, if they desired, depth as a repeated measure (repeated in space). The caveat is that inferences would only be applicable to the marshes studied because there was no true replication of the salinity gradient. So, extrapolation of conclusions to other marshes would have to be done cautiously. This is the approach this reviewer would take. Of course, one might posit that the main effects and interactions are obvious in Figure 4 and no stats are needed. Although this reviewer agrees that the effects are obvious, it's always better to have a rigorous statistical analysis supporting what appears intuitive. In summary, my recommendation is that an ANOVA of the data be performed and included in a revised manuscript. 14. Lines 119 -123: Comments about treatments being equal or different are not supported by statistical analyses, as discussed in comment 9. 15. Line 125: Insert "R2 = 0.58" before "p=1.086e-5". 16. Lines 127-129: Organic content was correlated with shear strength, yet organic matter data were not presented. The organic matter data should be in a graph or table. The probability of the relationship was close enough to 0.05 to be of interest. If the authors think the relationship is unimportant based on a probability of 0.059, they should state it was not significant. 17. Line 128: Instead of using the phrase "marginally significant", This reviewer suggests you simply state "...significant at p = 0.059". 18. Line 129: Water content is mentioned but no

data presented. Either delete water content in this line or say more about the relationship between it and organic content. 19. The authors statistically exlain variation in soil shear strength with salinity, aboveground biomass, belowground biomass, and bulk density. R2 values and regression graphs are provided for each of the predictor variables. This reviewer suggests the authors perform a stepwise multiple regression, similar to that done by Ford et al. (2016), to try to tease out the relative importance of each of these. Granted that some of the predictors might not be completely independent of each other, but this approach, especially when one varies the sequence by which the predictors enter the regression model, might help to better explain the important drivers of shear strength differences. 20. Lines 134-135: Sasser et al. 2018 also found that soil shear strength positively correlates with belowground biomass. Therefore, the present work confirms, rather than "extends", the concept. 21. Lines 138-139: This study did not directly measure "marsh erodibility", and hence the statement that "our results demonstrate that soil properties such as bulk density are also important drivers of marsh erodibility", although likely true, is not empirically demonstrated by this research. The sentence should be revised to reflect this. 22. Lines 139-140: This is an important finding. 23. Line 141: ANOVA would confirm this. 24. Lines 153-154: It is not clear to this reviewer if belowground biomass drives soil shear strength variability in all marsh types. Figure 5 shows that four data points drive the significant relationship between shear strength and belowground biomass. Are these data points from a single marsh type or study site? The relationship between belowground biomass and soil shear strength needs to be fleshed out more than just how it varies with marsh zone; marsh type should also be addressed in Figure 5 by using different shaped symbols. 25. Lines 154-155: It is stated "….and soil properties influence marsh edge shear strength (Fig. 6) for brackish and salt marshes". Figure 6 presents shear strength as a function of bulk density. It is clear that for salt marshes, bulk density is an important factor determining soil shear strength (p=0.002). However, the relationship is not significant for brackish marshes (p=0.318). Hence, the statement in lines 154-155 as written is incorrect. 26. Lines 160-162: The authors state that this study found that

salt marshes soils are generally stronger than fresh marsh soils. Howes et al (2010) concluded similarly. However, this reviewer believes we need to be cautious is making these broad statements about the importance of marsh type because soil strength may depend on the specific plant species that dominate the marsh, as well as the specific soil type in that marsh. For example, will a freshwater Peltandra marsh have the same soil strength as a freshwater Panicum marsh, and will they similarly differ from a Spartina alterniflora salt marsh? This reviewer submits the answer is likely – no, based on the work of Sasser et al. (2018), which showed large differences in shear strength both between marsh types and within marsh types. In addition, a freshwater marsh with a sapric organic soil is likely to have a different soil shear strength than a freshwater marsh with a mineral (entisol) soil, regardless of the dominant species. This reviewer suggests that the authors expand this section to include a discussion of these nuanced, but important, concepts. In summary, the differences seen in soil shear strength between salt and freshwater marshes are likely due to differences in dominant species, soil type, belowground biomass and structure, and hydrogeomorhic setting, all of which affect each other. 27. Lines 169-174: There is no discussion of the brackish marsh dominant and its root structure and anatomy. This should be added. 28. Lines 185-190: The important factor associated with biodiversity may not be biodiversity, per se, but rather the specific species contributing to the biodiversity. A monospecific stand of Spartina alterniflora may generate a greater soil shear strength than a diverse community of freshwater dicots with shallow and low-density roots. This concept may be similar to the importance of plant diversity to primary productivity and stability. It's not greater biodiversity that's important, but rather the species composition (or functional guilds) comprising the plant community. The authors should emphasize the potential importance of species composition. 29. Lines 196-197: This reviewer believes this statement is too simplistic. This reviewer agrees that all things being equal, saltwater intrusion alone would create stronger soils by promoting Spartina alterniflora and increasing bulk density via enhanced sedimentation. However, saltwater intrusion will be accompanied by increased water levels, assuming that sea-level rise is the driver of

**ESurfD**
[Figure]

**ESurfD**

Interactive
comment

the saltwater intrusion. Hence, coastal salt marshes will experience longer periods of inundation and higher water levels. Prolonged inundation will decrease root productivity and live belowground standing stock, resulting in a reduction in soil shear strength. Hence, the statement – "Although these changes will have a variety of ecological and geomorphic consequences, our work suggests that saltwater intrusion may be accompanied by stronger salt marsh soils that are less easily eroded." does not tell the whole story. 30. Figure 5. This reviewer suggests that the symbol style should differ by marsh type (salt, brackish, fresh) or by study site so that the reader can visualize the importance of marsh type in determining the relationship between belowground biomass and soil shear strength.

---

## Author Comment (AC1) · 31 Jan 2021

**Final Response**

Manuscript Identification: esurf-2020-58 Manuscript Title: Biophysical controls on marsh soil shear strength along an estuarine salinity gradient

Manuscript Authors: Megan N. Gillen, Tyler C. Messerschmidt, Matthew L. Kirwan

The authors would like to thank the two anonymous referees for their useful comments that helped improve this manuscript. We would also like to thank our associate editor for their help overseeing the manuscript review process. Below are explanations of our response to each reviewer's comments, with line numbers specified for reference where relevant. (RC1: Referee #1 comment; RC2: Referee #2 comment, AR: author response)

**Response to Comments from Anonymous Referee #1:**

**General Response:**

**RC1:** This paper investigates the influence of salinity and belowground biomass on soil shear strength in coastal wetlands of the Chesapeake Bay area. It is well-organized and clearly written. The data used and methods of analysis, are, in themselves, sound. However, I would like to suggest a few changes that would improve its clarity, and also point out what I consider to be weaknesses in their overall conclusion that, broadly speaking, tidal freshwater wetland soils have reduced soil shear strengths compared to their salt marsh counterparts.

**AR:** Thank you for your positive comments about our manuscript and for your suggestions towards improving our manuscript's clarity and overall conclusion. Below we have addressed your specific comments and suggested changes.

**Specific Comments:**

**RC1:** 1) I may have missed it earlier in the document, but I could not find in the text how the five sites were classified as either "fresh", "brackish", or "saline" until the caption for Figure 6. Prior to that figure, numerous references are made regarding these marsh types, but it is difficult to follow along with the data presented without knowing which sites were classified as which marsh types. Please specify this classification earlier in the paper (ideally in the methods/study site section).

**AR:** We have added "Goodwin Islands (salt), Catlett Islands (salt), Taskinas Creek (brackish), Sweet Hall Marsh (fresh), and the Pamunkey Indian Reservation (fresh)" to clarify our site classifications in the study site section (lines 64-65) in the manuscript text. The new table included in our manuscript (see below) also details the salinity regime for each study site, and is referenced in section *2.1 Study area and approach*.

**RC1:** 2) The bulk density values reported often seem quite high (many exceed the density of granite, commonly taken at 2.65 g/cm3). Are the bulk density values reported in the manuscript "wet" or "dry"? If they include water weight ("wet"), that would explain the high values.

**AR:** Thank you for bringing our attention to these bulk density values. These are dry bulk density weights, and we have clarified this in our methods section (line 107). We reviewed our raw data and original bulk density calculations, and realized there may have been errors in calculating the specific bulk density of samples below the upper 30-40 cm of some cores. We also discovered errors in some of previous soil property correlations due to excel graphically misrepresenting the data. We have redone this analysis entirely, both using only the upper 30 cm of each core and with all bulk density values, and found both times that bulk density no longer emerges as an important driver of shear strength. We now only include information regarding water content and organic content as they were the most significant drivers (see updated figure 6).

**RC1:** 3) The elevations measured at the Pamunkey site were much lower than those measured at the other sites. The authors explain this difference as possibly arising from the Pamunkey elevations being surveyed early spring prior to vegetation growth, while elevations at the other sites were measured well into the growing season. I suppose that makes sense to some degree, but the difference approaches 50 cm (through examination of Figure 1). That seems like a lot of elevation capital to attribute to a seasonal biomass cycle, and whether or not this site is considerably lower than the other site has fairly strong implications for interpretation of some of the differences seen in the data. It would be helpful if the authors could explain how before/during growing season differences could explain this much elevation difference in the marsh soil surface, which is presumably what was surveyed, and specifically what processes are at play here.

**AR:** This is a good point, and it is unfortunate that we were unable to measure the elevation profile at the Pamunkey site at the same time as the other sites. We originally chose our site locations by measuring the flooding depth during a propagating high tide. This appears to have been a successful approach for choosing sites with consistent elevations elsewhere in the estuary, though the Pamunkey certainly appears to be an outlier. Chesapeake Bay freshwater marshes are highly dynamic, and complete loss of vegetation does lead to thorough reworking of sediment, erosion, and reorganization of tidal channels (Pasternack and Brush, 1998). Nevertheless, it's impossible to know whether the ~50 cm difference in elevation represents seasonal erosion or a marsh that is indeed always lower in elevation than the others. Interestingly, the shear strength of the Pamunkey site is nearly identical to our other freshwater site (Sweet Hall) despite the large difference in elevation (*Sweet Hall*: 5.81 kPa on edge, 4.01 kPa in interior; *Pamunkey*: 4.98 kPa on edge, 6.75 kPa in interior). Therefore, our primary finding that shear strength increases with salinity is likely to hold regardless of the initial elevation of the Pamunkey site, and whether or not the Pamunkey site is included in the analysis. In

response to the reviewer comment, we have added text to the manuscript stating that we originally chose our site locations based on a consistent high tide flooding depth (observed while following a single propagating high tide up the estuary), but that the Pamunkey site is substantially lower in elevation than the others (lines 66-69, lines 90-91). We additionally now reference Pasternack and Brush in our discussion of potential seasonal erosion and subsidence related to loss of vegetation in colder months (line 94). We have also added text emphasizing that the shear strength is similar to our other freshwater site (Sweet Hall) despite its lower elevation, and substantially lower than the salt marsh sites, so that the general trend with salinity is unaffected:

"Interestingly, despite considerable differences in elevation (Fig. 1b), the shear strength values at the low salinity sites (Sweet Hall Marsh and the Pamunkey Indian Reservation) were similar (Fig. 2), ranging from 3.6-8.0 kPa for the upper 30 cm of the marsh interior with an overall average of 5.4 kPa. These shear strength values were also substantially lower than those reported from the high salinity sites (Fig. 3b), indicating that the general trend discovered between shear strength and salinity in the marsh interior is unaffected by the elevation discrepancies." (lines 137-142)

RC1: 4) Following up on comment (3) above, numerous studies exist that illustrate the importance of inundation in controlling belowground biomass. Many show hump-shaped responses, where high sites and low sites exhibit reduced biomass/productivity, and optimum production occurs at intermediate elevations. Others show monotonic production/biomass decreases with decreased elevation (or increased flooding). There is no mention of those dynamics to speak of in this manuscript. The authors should interpret their findings to some degree in the context of those studies. This is particularly the case given that their sites consistently show higher elevations at the "interior" locations than at the "edge" locations. Those differences are not trivial - mean interior elevations exceed their edge counter parts by 11cm, 14cm, 15cm, 11cm and 20cm for Pamunkey, Sweet Hall, Taskinas, Catlett, and Goodwin locations, respectively. I'm unfamiliar with how those elevation differences would translate into differences in hydroperiod in the Chesapeake Bay region, but in the microtidal, low gradient wetlands on the northern Gulf of Mexico coast, those elevation differences would easily translate into differences in flood duration that would exceed 30 % (e.g., an interior marsh at 0.25 m NAVD that was flooded at 40% of the time would be flooded around 70% or more of the time if its elevation were reduced by 0.15 m).

**AR:** While the focus of our manuscript is on the effects of salt water intrusion, not elevation, this is an interesting suggestion that warrants more attention. In our study, interior sites tend to have both higher elevations and higher shear strength values, suggesting the type of correlation noted by the reviewer is possible. However, the proposed mechanism (an optimum elevation for plant growth) is already tested directly with regressions between biomass and shear strength, and the results are subtle. Briefly, belowground biomass correlates with shear strength only in the interior of salt and brackish marshes. Nevertheless, we have attempted to follow reviewer advice and

made figures showing the relationship between elevation and shear strength, and the relationship between elevation and biomass. These figures show no evidence for an optimum elevation, and the type of discussion the reviewer suggests. However, we suggest that the general pattern that shear strength is maximized in the marsh interior is tied indirectly to elevation through its effect on soil properties including water content, belowground biomass, and organic content.

**Figure:** (a) Depth-averaged shear strength ( $\tau$ ) compared to elevation across all sites, grouped by edge and interior ( $R^2 = 0.54$ ; p = 0.02). Error bars represent standard error for depth-averaged shear strength measurements. (b) Depth-averaged belowground biomass compared to elevation across all sites, grouped by edge and interior ( $R^2 = 0.28$ ; p = 0.12). Error bars represent standard error for depth-averaged across all sites, grouped by edge and interior ( $R^2 = 0.28$ ; p = 0.12). Error bars represent standard error for depth-averaged belowground biomass measurements.

While our manuscript previously discussed these parameters, we have now added several sentences discussing elevation's influence over our measured parameters, and that in any case, these same parameters co-vary with each other and influence differences in elevation:

"Furthermore, other ecogeomorphic dynamics between biophysical parameters unexplored in this study may have considerable influence on marsh soil shear strength. For example, marsh elevation could have driven differences in biomass and soil properties within and across our sites through its relationship with inundation frequency and depth (Friedrichs and Perry, 2001; Kirwan and Guntenspergen, 2012; Morris et al., 2002). However, marsh elevation is also controlled by ecogeomorphic interactions between processes such as organic and mineral accretion, sediment trapping efficiency from aboveground stems, which in turn affect various biophysical parameters (Coleman and Kirwan, 2019; Donnelly and Bertness, 2001; Kirwan et al., 2016; Kirwan and Megonigal, 2013; Morris et al., 2002). Future work should consider the interplay between dominant species type, belowground biomass and root structure, soil type, and hydrogeomorphic setting and their effect on each other and, in turn, marsh soil shear strength." (lines 214-222)

Additionally, our new table (see below) includes information regarding the tidal range along the York River which could inform questions regarding flood duration and inundation depths. With regards to inundation controlling belowground biomass, we have added in our discussion section several phrases highlighting the importance of inundation as an interconnected, ecogeomorphic driver that may be driving differences in shear strength (lines 214-222 and response to RC2 comment #26 & #29).

**RC1:** 5) In Lines 151-153, enhanced nutrient loading at the edge sites relative to the interior sites is suggested as a possible explanation for why shear strength at the edge sites may be lower. However, according to figure 1, this distance is only on the order of 10 m. It does not seem reasonable to expect a meaningful reduction in nutrient concentrations over this length scale, but perhaps the authors can demonstrate otherwise with citations to support this claim.

**AR:** We have included additional citations which support our claim that enhanced nutrient loading may explain lower shear strength values at the marsh edge specifically (Johnson et al., 2016; Wigand et al., 2018). Johnson et al. states that tall form *Spartina alterniflora* responded to enrichment treatments while short form *S. alterniflora* and other high marsh plants (*Spartina patens, Distichlis spicata*) did not exhibit consistent reactions to nutrient enrichment. Although our distance between edge and interior sites is approximately 10 meters, for our salt marsh sites there is a shift in plant community from the tall form *S. alterniflora* dominated edge to mostly short form *S. alterniflora* in the interior. This vegetation gradient would then reflect the higher nutrient loading trends for edge sites found by Johnson et al., and additionally follows another conclusion that high marsh interior sites see lower enrichment loading rates. Nevertheless, eutrophication is simply part of a longer list of possible drivers for lower shear strength values at the marsh edge.

**RC1:** 6) The authors cite the Howes et al. (2010) paper that concludes that "salt marshes are more resistant to later edge erosion than freshwater marshes" (Lines 158-160, this manuscript). The Howes et al. (2010) paper identified Spartina patens as the dominant vegetation present in the low shear strength region of Breton Sound basin, Louisiana, that was so badly decimated by the shearing forces of Hurricane Katrina. They pointed out that although this species "has extensive rooting but of smaller diameter [than S. alterniflora]. The plant is less tolerant to anoxic soil conditions, which likely limits the root network to shallower depths" and use this logic to conclude explain why S. patens regions were sheared while S. alterniflora regions were largely intact after the storm. However, in lines 180-185 of this manuscript, the authors cite the presence of S. patens (and the co-occurring S. alterniflora) as the reason the salt marshes of the York River estuary exhibit the relatively high shear stresses, owing to their high productivity and their creation of dense networks of belowground biomass. It

is difficult to square these – Howes et al. (2010) on the one hand indicating that it is the limited development of the S. patens root network that contributes to a low shear stress, and this manuscript on the other hand citing the dense root network of S. patens in promoting high shear strength. It is important to consider, when citing Howes et al. (2010), that their "low salinity", "low shear strength" zones were vegetated almost exclusively by S. patens. In the present study, S. patens is identified as being in the high shear, high salinity zone.

**AR:** Thank you for bringing our attention to this discrepancy. We have removed the mention of *S. patens* from this section of the manuscript (line 229) as it assigned the presence of *S. patens* to only salt marshes, when globally it exists in marshes across a wider range of salinities. Additionally, there was actually little to no *S. patens* at our higher salinity sites we measured for this study. This is clarified through our inclusion of a new table highlighting the specific vegetation species present at each study site for edge and interior, as well as tidal range and salinity regime. We have included a reference to this new table in this section: "Salt marshes in the York River are dominated by *S. alterniflora* (Table 1)..." (lines 228-229)

**RC1:** 7) Given the rather large elevation gradients across each of the transects (see comment 3 above), it seems reasonable that vegetation species composition varies markedly across the transects (particularly between the "edge" and "interior" zones. Given that a paper cited frequently in this manuscript (Howes et al. 2010) attribute variations in shear strength to taxa-specific morphological differences in root structure, if transitions in species composition existed across the elevation gradients in the transects, within transect species composition could be responsible for some of the patterns observed in the manuscript. Can you speak differences in species composition not only between the transects, but within each transect as well?

**AR**: We have clarified the specific species type present at each study site for their respective edge and interior zones through our inclusion of a new table as described in the previous author comment. The primary difference in vegetation within transects for our salt marsh sites was the transition from tall to short form *S. alt*, which would have no taxa-specific morphological differences in root structure. At the brackish sites this same transition within our transect occurs, with the addition of *S. patens* and *Distichlis* in the marsh interior. While other species were present, we predominantly found *S. alt* roots in our marsh interior belowground biomass cores at Taskinas Creek so we wouldn't expect to see differences in species composition have a significant impact on shear strength at our brackish site. Lastly, for our freshwater marsh sites while there was variation in species present for edge and interior zones (see Table 1 below), *Peltandra virginica* dominated these sites in both zones and comprised the majority of the belowground biomass. Nevertheless, we have added several sentences to the discussion in response to Reviewer 2 that highlight the potential role that vegetation type plays in determining shear strength across sites (see response to RC2 #26).

**RC1:** 8) Similar to what was done for figure 6, could the regression in Figure 5 be separated by marsh type? For example, when breaking this regression down by site (see attached figure), a different picture emerges – biomass vs. shear strength is significant for Taskinas (the brackish site) and Goodwin (one of the saline sites), but insignificant for the remaining three sites. Beyond simple reductions in sample sizes and degrees of freedom, are there other explanations for why this relation may be significant at some of the sites (or marsh types) but insignificant at other sites?

**AR:** We have conducted further analysis on the belowground biomass regression as shown in Figure 5 to reflect correlations based on salinity regime in the marsh interior. We found that relationships were only significant for the salt and brackish marsh sites. However, this may be due to the overall low amount of belowground biomass that exists in freshwater marshes, which would support the relationship we see in the marsh interior across all 5 sites. For this reason, we maintain that belowground biomass drives shear strength in the marsh interior across all sites, regardless of salinity. We have included text about this point in our discussion section: "The relationship between belowground biomass and shear strength was not significant at our freshwater marsh sites (Fig. 5b). However, this is likely due to the overall lower amount of belowground biomass present in York River freshwater marshes that would not produce a significant linear relationship compared to the range of biomass values found in our salt and brackish sites. Therefore, we maintain that differences between belowground biomass drive shear strength values in the marsh interior regardless of salinity, where low biomass values relate to low shear strength values both within a soil profile and across different marsh types." (lines 168-173). We have also further revised Figure 5 entirely (see below), dividing into two panels: the first depicting the same data as before, but grouped by salinity type (indicated by color), the second depicting only the interior data and with correlations for each marsh salinity type.

**Response to Comments from Anonymous Referee #2:**

**General Response:**

**RC2:** This reviewer believes that the data presented in the manuscript is of value and should be publishable. However, there are numerous problems (see below) that first need to be addressed.

**AR:** Thank you for your positive comments regarding our manuscript and for your suggestions towards improving our manuscript's clarity and overall conclusion. Below we have addressed your specific comments and suggested changes.

**Specific Comments:**

**RC2:** 1. Line 11: "erodibility" was no directly measured in this study. Insert "potential" between before "marsh erodibility".

AR: We have added "potential" before marsh erodibility in line 11.

RC2: 2. Line 12: Insert "likely" before driven.

AR: We have added "likely" before driven to line 12 (now line 13).

**RC2:** 3. "rooting structure" was not measured in this study. Hence, the need for comment.

**AR:** We have removed the mention of "rooting structure" from line 13 in the abstract.

**RC2:** 4. Lines 15-16: The last sentence in the abstract is misleading and not supported by the data presented in the manuscript. Based on the data presented in the manuscript, one can only state that the freshwater marshes investigated had weaker soils than the salt marshes investigated. Extrapolating to salt marshes and fresh marshes universally goes far beyond this specific study, and, based on Table 3 in Sasser et al. (2018), is an unsupported generalization.

**AR:** We have clarified this last sentence by adding "York River" before "freshwater marshes" (line 15) and "these" before "freshwater marshes" (line 16) to illustrate that our findings relate specifically to marshes along the York River in southeastern Virginia.

**RC2:** 5. Line 65: the statement is made that there "were consistent elevations across all study sites" and Fig 1b. is referenced. However, when this reviewer looks at Fig. 1b, one sees elevations that differ both among the marshes and within a marsh. Hence, this statement appears incorrect and may need qualification.

**AR:** The authors felt additional information was needed regarding our sampling to clarify our selection criteria. Hence, we have added "Within these overall sites, we chose sampling locations during a July 2018 survey cruise that followed the propagation of high tide along the York River. Sampling locations were selected based on similar flooding depths at high tide to maintain consistency in inundation depths and along tidal creeks 5-10 m wide, with marsh widths beyond 20 meters." (lines 66-69)

**RC2:** 6. Lines 65-68: The reviewer suggests rewriting these lines as follows: We collected samples from two zones within each marsh: (1) the tidal channel marsh edge located between the tidal channel and any levee (1 m from edge) and (2) the interior marsh located at a measured distance of 10-12 m away from the edge site (Fig. 1b).

**AR:** We have rewritten lines 65-68 (now 69-71) as "We collected samples from two zones within each marsh: (1) the tidal channel marsh edge located between the tidal channel and any levee (1 m from edge) and (2) the interior marsh located at a measured distance of 10-12 m away from the edge site (Fig. 1b)." as suggested.

**RC2:** 7. Lines 78-79: Provide information on the spatial distribution of the 10 replicated profiles. Were they within a 1-meter radius, for example, or where they distributed over a 10 m distance parallel to the shoreline, for example?

**AR:** We have updated lines 78-79 (now lines 83-85) to include information regarding the spatial distribution of our shear strength samples "Ten replicate profiles were taken within 1 meter of each other over a 10 meter distance parallel to the shoreline per marsh location at each study site..."

**RC2:** 8. Lines 83-84: It is stated that "All sites were located at similar elevations....."., However, Fig. 1b shows that, for example, the Catlett study site had dissimilar elevations within the edge zone. Maybe what would help is to provide the range in elevations for the specific zones in each marsh where shear strength was measured. A table might also work. The bottom line is that the authors do not adequately convince this reader that elevations were "similar".

**AR:** Thank you for your comment. We have taken your suggestion and provided ranges of elevation for each marsh zone. The updated manuscript now reads "Edge elevations ranged between 0.1 - 0.5 m and 0.3 - 0.6 m in the marsh interior for all sites except for the Pamunkey Indian Reservation, which was lower in elevation than the other sites." (lines 89-91)

**RC2:** 9. For both above- and belowground sampling, the spatial distribution of the three replicates should be provided. Were they within a meter of each other or 10 meters of each other?

**AR:** We have provided information regarding the spatial distribution of above-ground samples (line 98: "we collected...aboveground stem clip plots (25 cm x 25 cm) located within 1-2 meters of each other per marsh zone...") and belowground samples (line 101-102): "belowground biomass soil cores (15 cm diameter, 50-70 cm depth) were collected within the aboveground plots after destructive harvest at each location within sites...").

**RC2:** 10. Generally, soil shear strength is very much dependent on the plant species that occur where measurements are taken. Knowing the dominant community type does not provide enough information. Hence, within the Methods or Results, the authors should specify the specific plant species present where the shear strength vanes were inserted. In the salt marsh, of course, it is likely to be Spartina alterniflora, but in the brackish marsh, it could be any number of species (e.g., S. patens, S. cynosuroides, or Distichlis, etc.), and in the fresher marshes it could be a mixture of species or one dominant like Peltandra. This information is needed to better interpret the shear strength results.

**AR:** Thank you for your suggestion. We have clarified the species type present at each site with an inclusion of a new table (see below) that lists species type present for both edge and interior zones, the respective tidal range, and salinity regime for our five study sites.

**RC2:** 11. Lines 57-101: This reviewer is surprised that soil textural components were not measured. Are sand, silt, and clay data available for these study sites?

**AR:** While the authors did not measure soil textural components for this study, we know from the literature and observation that our polyhaline sites are overall sandier in composition due to their proximity to the mouth of the York River. Our freshwater and brackish sites are comprised of mix of sand and fine-grained muds. We have now included grain size descriptions in section *2.1 Study area and approach*: "Grain size on the river bed shifts from predominantly sand in the lower York River to a mud-sand mix in the middle and upper reaches of the estuary (Gillett and Schaffner, 2009)." and cited Gillett and Schaffner 2009 (see their figure 10) to clarify this missing information in the manuscript (lines 56-57).

**RC2:** 12. A Statistical Analysis sub-section is absent from the Methods section. The authors should describe the experimental design of their study and the statistical methods used to test significance. In this regard, this reviewer finds it difficult to understand why the authors did not use an Analysis of Variance approach to identify significant differences among study sites and between zones, as well as their interaction, which they graphically present in Figure 4 and discuss in lines 119-123. See below for more on this.

**AR:** Thank you for your comment. We have added a sub-section detailing our methodology for the statistical analysis portion of this study after section *2.2 Measurements of shear strength, vegetation, and soil properties* entitled *2.3 Statistical analysis* (lines 112-121). We have also included information regarding the newly included ANOVA analysis (lines 117-121). See the author response to the next comment for a more detailed description on our newly incorporated ANOVA analysis. Below is the newly included statistical analysis subsection included in our methods:

"We conducted all statistical analysis in Microsoft Excel. Replicate measurements were averaged together to create composite profiles for shear strength and biomass data. We employed simple linear regression analysis to determine significant correlations between shear strength and biophysical drivers. R2 and p-values were calculated for each relationship using the regression tool from the Microsoft Excel Analysis ToolPak. In linear regression analyses broken down by salinity type, we simply grouped together data points from study sites with the same salinity regime (Table 1). To test for significant spatial differences in shear strength, we used a two-way Analysis of Variance (ANOVA) with marsh type (i.e., salt, brackish, and fresh) and marsh zone as the primary treatments. Shear strength values were averaged at concurrent depths for (1) Goodwin Islands and

Catlett Islands and (2) Sweet Hall Marsh and Pamunkey Indian Reservation to create composite profiles for salt and freshwater marshes, respectively, for the ANOVA." (lines 113-121)

**RC2:** 13. Lines 119-123: Figure 4, which is discussed in lines 119-123, may be the most important results of the study. Yet, it received no statistical analysis. Although one might argue that the study is pseudoreplicated, given that the York River salinity gradient was not replicated i.e., two or more rivers), one can still do a 2-way factorial ANOVA with study site (the five marshes) and marsh zone (edge and interior) as the main treatments. The marsh zones should be nested within study sites. The authors could even include, if they desired, depth as a repeated measure (repeated in space). The caveat is that inferences would only be applicable to the marshes studied because there was no true replication of the salinity gradient. So, extrapolation of conclusions to other marshes would have to be done cautiously. This is the approach this reviewer would take. Of course, one might posit that the main effects and interactions are obvious in Figure 4 and no stats are needed. Although this reviewer agrees that the effects are obvious, it's always better to have a rigorous statistical analysis supporting what appears intuitive. In summary, my recommendation is that an ANOVA of the data be performed and included in a revised manuscript.

**AR:** Thank you for your comment. We have conducted a 2-way ANOVA as suggested with study site and marsh zone as the main treatments, and found statistically significant differences in shear strength between study sites and the marsh edge and interior. We also repeated this analysis grouping sites by salinity type (salt = Goodwin and Catlett, brackish = Taskinas, fresh = Sweet Hall and Pamunkey) and also found statistically significant differences. We have used the results from this second analysis in our manuscript as it corresponds with the data presented Figure 4. We have included a description of our ANOVA analysis (see previous comment) and our ANOVA results (lines 145-146). See below for our ANOVA results tables:

| SUMMARY       | goodwin    | catlett    | taskinas   | sweet hall | pamunkey   | Total      |
|---------------|------------|------------|------------|------------|------------|------------|
| edge          |            |            |            |            |            |            |
| Count         | 30         | 30         | 30         | 30         | 30         | 150        |
| Sum           | 110        | 195.5      | 134.25     | 175        | 149.375    | 764.125    |
| Average       | 3.66666667 | 6.51666667 | 4.475      | 5.83333333 | 4.97916667 | 5.09416667 |
| Variance      | 32.9195402 | 15.829454  | 4.80646552 | 2.6566092  | 17.1671157 | 15.2894162 |
| interior      |            |            |            |            |            |            |
| Count         | 30         | 30         | 30         | 30         | 30         | 150        |
| Sum           | 495        | 614        | 498        | 120.375    | 202.375    | 1929.75    |
| Average       | 16.5       | 20.4666667 | 16.6       | 4.0125     | 6.74583333 | 12.865     |
| Variance      | 238.724138 | 46.6022989 | 71.6394397 | 2.94434267 | 6.54146911 | 111.744237 |
| Total         |            |            |            |            |            |            |
| Count         | 60         | 60         | 60         | 60         | 60         |            |
| Sum           | 605        | 809.5      | 632.25     | 295.375    | 351.75     |            |
| Average       | 10.0833333 | 13.4916667 | 10.5375    | 4.92291667 | 5.8625     |            |
| Variance      | 175.391243 | 80.1620056 | 74.9519597 | 3.5959172  | 12.446875  |            |
| ANOVA         |            |            |            |            |            |            |
| rce of Variat | SS         | df         | MS         | F          | P-value    | F crit     |
| Sample        | 4528.9388  | 1          | 4528.9388  | 102.970007 | 6.648E-21  | 3.87372423 |
| Columns       | 3010.62104 | 4          | 752.65526  | 17.112379  | 1.2983E-12 | 2.40277496 |
| Interaction   | 3162.29792 | 4          | 790.574479 | 17.9745108 | 3.3653E-13 | 2.40277496 |
| Within        | 12755.0953 | 290        | 43.9830873 |            |            |            |
| Total         | 23456.9531 | 299        |            |            |            |            |

**(1) ANOVA with study site and marsh zone as main treatments**

| SUMMARY             | salt       | brackish   | fresh      | Total      |            |            |
|---------------------|------------|------------|------------|------------|------------|------------|
| edge                |            |            |            |            |            |            |
| Count               | 30         | 30         | 30         | 90         |            |            |
| Sum                 | 152.75     | 134.25     | 162.1875   | 449.1875   |            |            |
| Average             | 5.09166667 | 4.475      | 5.40625    | 4.99097222 |            |            |
| Variance            | 13.463829  | 4.80646552 | 7.07482489 | 8.40981049 |            |            |
| interior            |            |            |            |            |            |            |
| Count               | 30         | 30         | 30         | 90         |            |            |
| Sum                 | 554.5      | 498        | 161.375    | 1213.875   |            |            |
| Average             | 18.4833333 | 16.6       | 5.37916667 | 13.4875    |            |            |
| Variance            | 80.2712644 | 71.6394397 | 3.32298851 | 84.4214537 |            |            |
| Total               |            |            |            |            |            |            |
| Count               | 60         | 60         | 60         |            |            |            |
| Sum                 | 707.25     | 632.25     | 323.5625   |            |            |            |
| Average             | 11.7875    | 10.5375    | 5.39270833 |            |            |            |
| Variance            | 91.6672669 | 74.9519597 | 5.11097612 |            |            |            |
|                     |            |            |            |            |            |            |
| Source of Variation | .55        | df         | MS         | F          | P-value    | F crit     |
| Sample              | 3248.59429 | 1          | 3248.59429 | 107.939384 | 5.6218E-20 | 3.8954579  |
| Columns             | 1378.49484 | 2          | 689.247418 | 22.9012721 | 1.4823E-09 | 3.04790648 |
| Interaction         | 1646.70213 | 2          | 823.351063 | 27.3570655 | 4.6698E-11 | 3.04790648 |
| Within              | 5236.78555 | 174        | 30.0964687 |            |            |            |
| Total               | 11510.5768 | 179        |            |            |            |            |

(2) ANOVA with marsh type (i.e., salt, brackish, fresh) and marsh zone as main treatments

**RC2:** 14. Lines 119 -123: Comments about treatments being equal or different are not supported by statistical analyses, as discussed in comment 9.

**AR:** We have added information regarding the ANOVA analysis to support our comments regarding the magnitude of shear strength differences between edge and interior sites (see comments above).

**RC2:** 15. Line 125: Insert "R2 = 0.58" before "p=1.086e-5".

**AR:** We have inserted " $R^2 = 0.58$ " before "p = 1.086e-5" in the parentheses in line 125 (now line 151) of the updated manuscript.

**RC2:** 16. Lines 127-129: Organic content was correlated with shear strength, yet organic matter data were not presented. The organic matter data should be in a graph or table. The probability of the relationship was close enough to 0.05 to be of interest. If the authors think the relationship is unimportant based on a probability of 0.059, they should state it was not significant.

**AR:** Thank you for your comment. After reviewing our soil property data, we discovered some mistakes where excel altered the data graphically. The p-value for organic content is still correct (p = 0.059), however the new R2 value indicates little correlation between organic content and shear strength ( $R^2 = 0.09$ ). Because of this we have revised the discussion of soil properties in this section with our updated, correct analysis. Lines 153-156) now read: "Water content was significantly correlated with edge shear strength values ( $R^2 = 0.76$ , p = 5.717e-14) (Fig. 6a). However, other properties that co-varied with water content were also important, including the relationship between organic content and shear strength at edge sites in salt marshes (Fig. 6b)." We have also revised Figure 6 (see below) to include another panel with a scatter plot depicting the organic content data against concurrent shear strength values at edge sites, separated by salinity type (as organic content is significantly correlated with shear strength at salt marsh edge sites).

**RC2:** 17. Line 128: Instead of using the phrase "marginally significant", This reviewer suggests you simply state "...significant at p = 0.059".

**AR:** We have removed this phrase with our updated soil properties analysis results (see response to comment above).

**RC2:** 18. Line 129: Water content is mentioned but no data presented. Either delete water content in this line or say more about the relationship between it and organic content.

**AR:** We now directly address the relationship between water content and shear strength after revising our soil properties analyses (see comments above). Water content data is also now depicted in the revised Figure 6, separated by edge and interior (Fig. 6a).

**RC2:** 19. The authors statistically explain variation in soil shear strength with salinity, aboveground biomass, belowground biomass, and bulk density. R2 values and regression graphs are provided for each of the predictor variables. This reviewer suggests the authors perform a stepwise multiple regression, similar to that done by Ford et al. (2016), to try to tease out the relative importance of each of these. Granted that some of the predictors might not be completely independent of each other, but this approach, especially when one varies the sequence by which the predictors enter the regression model, might help to better explain the important drivers of shear strength differences.

**AR:** Thank you for your comment. Prior to our submission of this manuscript we had attempted both a stepwise multiple regression and simple multiple linear regression analysis. However, we encountered several issues with parameter dimensions. In order to match the varying number of replicates between different measured biophysical parameters and shear strength (biomass = 3 replicates, soil properties = 1 replicate, shear strength = 10 replicates), to generate the model we had to create a single, averaged data point per depth at each study site (i.e., one averaged bulk density measurement for Goodwin edge at 17.5 cm in the soil profile). We also only used values from the top 50 cm of the soil profile to further match parameter spaces and maintain consistency across all five sites. After all this data preparation, we were left with a total of 50 rows of data to perform our regression, 25 per marsh zone when we separated the analysis by edge and interior (since we only found a significant relationship between shear strength and salinity in the marsh interior). The authors felt that this did not constitute enough data to most accurately make a model, which was also reflected in our calculated  $R^2$  which did not exceed 0.53 in all our model setups.

This approach also forced us to exclude aboveground biomass as there was no depth variable associated with this biomass property. We did attempt a depth averaged analyses (similar to what we did comparing shear strength with salinity), however there were too few parameters to accurately make a model and we generated an unrealistic R2 value of 0.96 (based on 10 rows of data). This method also excluded important trends with depth that existed for our biophysical parameters (i.e., a decrease in belowground biomass with depth).

**RC2:** 20. Lines 134-135: Sasser et al. 2018 also found that soil shear strength positively correlates with belowground biomass. Therefore, the present work confirms, rather than "extends", the concept.

**AR:** Thank you for your comment. We have rephrased this statement to "our work confirms..." (line 160).

**RC2:** 21. Lines 138-139: This study did not directly measure "marsh erodibility", and hence the statement that "our results demonstrate that soil properties such as bulk density are also important drivers of marsh erodibility", although likely true, is not

empirically demonstrated by this research. The sentence should be revised to reflect this.

**AR:** We have added "potential" before "marsh erodibility" to reflect the caveat highlighted in this comment, similar to the RC2 #1 asking for this change in our abstract (line 165).

RC2: 22. Lines 139-140: This is an important finding.

**AR:** Thank you for your comment.

RC2: 23. Line 141: ANOVA would confirm this.

**AR:** ANOVA has confirmed that the marsh interior has higher soil shear strength at the marsh edge for our salt and brackish marsh sites (see response to RC2 comment #13).

**RC2:** 24. Lines 153-154: It is not clear to this reviewer if belowground biomass drives soil shear strength variability in all marsh types. Figure 5 shows that four data points drive the significant relationship between shear strength and belowground biomass. Are these data points from a single marsh type or study site? The relationship between belowground biomass and soil shear strength needs to be fleshed out more than just how it varies with marsh zone; marsh type should also be addressed in Figure 5 by using different shaped symbols.

**AR:** We have revised Figure 5 to differentiate between different marsh type (i.e. salt, brackish, and fresh) by color. We also conducted further basic linear regression analysis on belowground biomass and shear strength grouped by salinity type, and found only significant relationships in salt and brackish marshes. However, this is likely due to the overall lower amount of belowground biomass present in freshwater marshes that would not produce a significant linear relationship compared to the range of biomass values found in salt and brackish sites (no live biomass typically found below 30 cm in the soil profile). Therefore, we maintain that differences between belowground biomass drive shear strength in the marsh interior regardless of salinity, where low biomass values warrant low shear strength values both within a soil profile and across different marsh types. We have added this point to our discussion section in the manuscript as well:

"The relationship between belowground biomass and shear strength was not significant at our freshwater marsh sites (Fig. 5b). However, this is likely due to the overall lower amount of belowground biomass present in York River freshwater marshes that would not produce a significant linear relationship compared to the range of biomass values found in our salt and brackish sites. Therefore, we maintain that differences between belowground biomass drive shear strength values in the marsh interior regardless of salinity, where low biomass values relate to low shear strength values both within a soil profile and across different marsh types." (lines 168-173). **RC2:** 25. Lines 154-155: It is stated "....and soil properties influence marsh edge shear strength (Fig. 6) for brackish and salt marshes". Figure 6 presents shear strength as a function of bulk density. It is clear that for salt marshes, bulk density is an important factor determining soil shear strength (p=0.002). However, the relationship is not significant for brackish marshes (p=0.318). Hence, the statement in lines 154-155 as written is incorrect.

**AR:** We have removed the mention of both salt and brackish marshes in our revised manuscript (see response to RC2 comment #16) as we found that bulk density is no longer correlated with shear strength (see response to RC1 comment #2), and because water content is significantly correlated with edge shear strength values across all study sites.

RC2: 26. Lines 160-162: The authors state that this study found that salt marshes soils are generally stronger than fresh marsh soils. Howes et al (2010) concluded similarly. However, this reviewer believes we need to be cautious is making these broad statements about the importance of marsh type because soil strength may depend on the specific plant species that dominate the marsh, as well as the specific soil type in that marsh. For example, will a freshwater Peltandra marsh have the same soil strength as a freshwater Panicum marsh, and will they similarly differ from a Spartina alterniflora salt marsh? This reviewer submits the answer is likely - no, based on the work of Sasser et al. (2018), which showed large differences in shear strength both between marsh types and within marsh types. In addition, a freshwater marsh with a sapric organic soil is likely to have a different soil shear strength than a freshwater marsh with a mineral (entisol) soil, regardless of the dominant species. This reviewer suggests that the authors expand this section to include a discussion of these nuanced, but important, concepts. In summary, the differences seen in soil shear strength between salt and freshwater marshes are likely due to differences in dominant species, soil type, belowground biomass and structure, and hydrogeomorhic setting, all of which affect each other.

**AR:** Thank you for your comment. We recognize the importance of other biophysical parameters that were not examined in this study, especially species type, that may have considerable influence over shear strength. To emphasize the potential importance of species type, we have added the follow sentence in this section to lines 195-198: "This relationship between salinity and shear strength may be also species dependent—while our *Peltandra virginica*-dominated freshwater marsh sites had the weakest soils, previous work shows other freshwater grass species such as *Panicum hemitomon* with relatively high shear strength values (Sasser et al., 2018)." We have also added further mention to the importance of species type in our discussion paragraph on root structure and geometry: "Investigations into root structure and geometry may also clarify how species type influences the relationship between belowground biomass and marsh soil shear strength (Sasser et al., 2018), and should be incorporated into future work." (lines

207-209). Lastly, to highlight other ecogeomorphic dynamics that were not explored in this study, we have created a separate paragraph rearranging sentences previously comprising the end of the salinity discussion section:

"Interestingly, we found no relationship between salinity and soil shear strength at the marsh edge, where erosion would actually occur (Fig. 3a). Although this finding warrants more attention, we suggest that processes associated with a more dynamic marsh edge (e.g., sediment deposition, erosion, and resuspension) obscure patterns that would otherwise be evident. Furthermore, other ecogeomorphic dynamics between biophysical parameters unexplored in this study may have considerable influence on marsh soil shear strength. For example, marsh elevation could have driven differences in biomass and soil properties within and across our study sites through its relationship with inundation frequency and depth (Friedrichs and Perry, 2001; Kirwan and Guntenspergen, 2012; Morris et al., 2002). However, marsh elevation is also controlled by ecogeomorphic interactions between processes such as organic and mineral accretion, sediment trapping efficiency from aboveground stems, which in turn affect various biophysical parameters (Coleman and Kirwan, 2019; Donnelly and Bertness, 2001; Kirwan et al., 2016; Kirwan and Megonigal, 2013; Morris et al., 2002). Future work should consider the interplay between dominant species type, belowground biomass and root structure, soil type, and hydrogeomorphic setting and their effect on each other and, in turn, marsh soil shear strength." (lines 211-219)

**RC2:** 27. Lines 169-174: There is no discussion of the brackish marsh dominant and its root structure and anatomy. This should be added.

**AR:** The brackish marsh dominant species is the same as the salt marshes (*S. alt*). We have explained this in our new table, and clarified this point in this section: "In the salt and brackish marshes, *Spartina*-dominated systems..." (line 203)

**RC2:** 28. Lines 185-190: The important factor associated with biodiversity may not be biodiversity, per se, but rather the specific species contributing to the biodiversity. A monospecific stand of Spartina alterniflora may generate a greater soil shear strength than a diverse community of freshwater dicots with shallow and low-density roots. This concept may be similar to the importance of plant diversity to primary productivity and stability. It's not greater biodiversity that's important, but rather the species composition (or functional guilds) comprising the plant community. The authors should emphasize the potential importance of species composition.

**AR:** We have highlighted the potential importance of plant community on shear strength differences in this section by adding the following at the end of the paragraph: "This difference in plant community between the York River and Essex/Morecambe Bay marshes additionally underscores the potential importance of species composition in driving variation in shear strength (Sasser et al., 2018), and should be investigated in tandem with biodiversity in future studies." (line 238-240)

**RC2:** 29. Lines 196-197: This reviewer believes this statement is too simplistic. This reviewer agrees that all things being equal, saltwater intrusion alone would create stronger soils by promoting Spartina alterniflora and increasing bulk density via enhanced sedimentation. However, saltwater intrusion will be accompanied by increased water levels, assuming that sea-level rise is the driver of the saltwater intrusion. Hence, coastal salt marshes will experience longer periods of inundation and higher water levels. Prolonged inundation will decrease root productivity and live belowground standing stock, resulting in a reduction in soil shear strength. Hence, the statement – "Although these changes will have a variety of ecological and geomorphic consequences, our work suggests that saltwater intrusion may be accompanied by stronger salt marsh soils that are less easily eroded." does not tell the whole story.

**AR:** We have revised lines 243-248 to reflect additional consequences of saltwater intrusion (changes highlighted in bold): "While accelerated rates of sea-level rise could enhance wave erosion (Mariotti and Fagherazzi, 2010), **increase inundation of marshes**, and threaten their survival (Kirwan and Megonigal, 2013), sea-level rise also leads to changes in vegetation type and productivity (Donnelly and Bertness, 2001; Kirwan et al., 2009; Morris et al., 2002). Although these changes will have a variety of ecological and geomorphic consequences, our work suggests that saltwater intrusion **alone** could be accompanied by stronger salt marsh soils that are less easily eroded."

**RC2:** 30. Figure 5. This reviewer suggests that the symbol style should differ by marsh type (salt, brackish, fresh) or by study site so that the reader can visualize the importance of marsh type in determining the relationship between belowground biomass and soil shear strength.

**AR:** We have grouped the marsh types by color (salt = blue, brackish = purple, fresh = green) in the updated Figure 5 (see below). Additionally, we have added another panel to figure 5 depicting only the marsh interior points again grouped by salinity, with relationships illustrated for each marsh type.

**Additional manuscript revisions:**

- Updated all p-values to 3 significant figures.
- Line 64: Added "Table 1" in parentheses with "Fig. 1a" to reference our new table including site information.
- Line 71: Changed "10-12 m" to "10-15 m" after reviewing elevation data in response to RC1 comment #4.
- Lines 98 & 99: Removed "-" in "above-ground"
- Lines 109-110: Revised "Cores were typically sectioned into 1 cm segments for the top 30 cm and at varying 2-5 cm intervals for the bottom 70 cm."
- Line 149: Changed "3.1" to "3.2"
- Line 164-165: Changed "bulk density" to "water content and organic content"

- Line 229: Added mention of Table 1 after "S. alterniflora"
- Line 234: Changed "U.K." to "Essex/Morecambe Bay"
- Line 235-236: added "previous work in the" before "U.K."
- Figure 1: Revised where are interior zones were located after reviewing the elevation data in response to RC1 comment #4
- Added (Sisson et al., 1997) as a citation to Table 1 and in the references section.
- Figure 2a: Changed "Catalett" to "Catlett" in legend
- Figure 2, Figure 3, Figure 4: italicized tau symbol along respective axes
- Figure 7: Added "and brackish" to caption after "salt" to indicated *S. alt* is also our dominant brackish marsh species.

**References:**

Coleman, D. J. and Kirwan, M. L.: The effect of a small vegetation dieback event on salt marsh sediment transport, Earth Surf. Process. Landf., 44(4), 944–952, https://doi.org/10.1002/esp.4547, 2019.

Donnelly, J. P. and Bertness, M. D.: Rapid shoreward encroachment of salt marsh cordgrass in response to accelerated sea-level rise, Proc. Natl. Acad. Sci., 98(25), 14218–14223, https://doi.org/10.1073/pnas.251209298, 2001.

Friedrichs, C. T. and Perry, J. E.: Tidal Salt Marsh Morphodynamics: A Synthesis, J. Coast. Res., 7–37, 2001.

Gillett, D. J. and Schaffner, L. C.: Benthos of the York River, J. Coast. Res., 10057, 80–98, https://doi.org/10.2112/1551-5036-57.sp1.80, 2009.

Johnson, D. S., Warren, R. S., Deegan, L. A. and Mozdzer, T. J.: Saltmarsh plant responses to eutrophication, Ecol. Appl., 26(8), 2649–2661, https://doi.org/10.1002/eap.1402, 2016.

Kirwan, M. L. and Guntenspergen, G. R.: Feedbacks between inundation, root production, and shoot growth in a rapidly submerging brackish marsh, J. Ecol., 100(3), 764–770, https://doi.org/10.1111/j.1365-2745.2012.01957.x, 2012.

Kirwan, M. L. and Megonigal, J. P.: Tidal wetland stability in the face of human impacts and sea-level rise, Nature, 504(7478), 53–60, https://doi.org/10.1038/nature12856, 2013.

Kirwan, M. L., Guntenspergen, G. R. and Morris, J. T.: Latitudinal trends in Spartina alterniflora productivity and the response of coastal marshes to global change, Glob. Change Biol., 15(8), 1982–1989, https://doi.org/10.1111/j.1365-2486.2008.01834.x, 2009.

Kirwan, M. L., Temmerman, S., Skeehan, E. E., Guntenspergen, G. R. and Fagherazzi, S.: Overestimation of marsh vulnerability to sea level rise, Nat. Clim. Change, 6(3), 253–260, https://doi.org/10.1038/nclimate2909, 2016.

Mariotti, G. and Fagherazzi, S.: A numerical model for the coupled long-term evolution of salt marshes and tidal flats, J. Geophys. Res., 115(F1), F01004, https://doi.org/10.1029/2009JF001326, 2010.

Morris, J. T., Sundareshwar, P. V., Nietch, C. T., Kjerfve, B. and Cahoon, D. R.: Responses of Coastal Wetlands to Rising Sea Level, Ecology, 83(10), 2869–2877, https://doi.org/10.2307/3072022, 2002.

Pasternack, G. B. and Brush, G. S.: Sedimentation cycles in a river-mouth tidal freshwater marsh, Estuaries, 21(3), 407–415, https://doi.org/10.2307/1352839, 1998.

Sasser, C. E., Evers-Hebert, E., Holm, G. O., Milan, B., Sasser, J. B., Peterson, E. F. and DeLaune, R. D.: Relationships of Marsh Soil Strength to Belowground Vegetation Biomass in Louisiana Coastal Marshes, Wetlands, 38(2), 401–409, https://doi.org/10.1007/s13157-017-0977-2, 2018.

Sisson, G. M., Shen, J., Kim, S. and Boon, J.D.: VIMS Three-Dimensional Hydrodynamic-Eutrophication Model (HEM-3D): Application of the Hydrodynamic Model to the York River System, Special Reports in Applied Marine Science and Ocean Engineering (SRAMSOE), Virginia Institute of Marine Science, College of William & Mary. https://publish.wm.edu/reports/1045, last access: 7 January 2021, 1997.

Wigand, C., Watson, E. B., Martin, R., Johnson, D. S., Warren, R. S., Hanson, A., Davey, E., Johnson, R. and Deegan, L.: Discontinuities in soil strength contribute to destabilization of nutrient-enriched creeks, Ecosphere, 9(8), e02329, https://doi.org/10.1002/ecs2.2329, 2018.

---

## Author Response (AR2)

**Minor Revisions**

**Manuscript Identification:** esurf-2020-58
**Manuscript Title:** Biophysical controls on marsh soil shear strength along an estuarine salinity gradient
**Manuscript Authors:** Megan N. Gillen, Tyler C. Messerschmidt, Matthew L. Kirwan

The authors would like to thank the two anonymous referees for their useful comments that helped improve this manuscript. We would also like to thank our associate editor for their help overseeing the manuscript review process. Below are explanations of our response to reviewer 2's comments with regards to minor revisions, with line numbers specified for reference where relevant. (RC2: Referee #2 comment, AR: author response)

**Response to Comments from Anonymous Referee #2:**

**RC2:** 1. Line 144: "Interior sites yielded higher values of shear strength than edge sites in the brackish and the salt marshes (Fig. 4)."
Reviewer Comment: Does the following statement signify a significant interaction between marsh type and marsh zone? It would appear so, but it should be specifically stated.

**AR:** We have revised this section of the results and added: "This result from the ANOVA indicates that the effect of marsh zone on shear strength varied with marsh type" (line 146-147) to clarify the significant interaction between marsh type and marsh zone.

**RC2:** 2. Lines 144-146: "While shear strength values appear nearly equal between the edge and interior sites for freshwater marshes (Fig. 4), the ANOVA test showed significant differences in shear strength between marsh edge and interior across all marsh types (p = 5.62e-20)."
Reviewer Comment: This sentence appears to state that the main effect of marsh zone (edge versus interior) is significant. However, because there is a significant interaction between the main effects of marsh type and zone (as shown in the ANOVA tables provided as responses to the first review), the main effects are little value, i.e., the effect of marsh zone on shear strength varies with marsh type. Hence, there is a need to discuss the results with respect to the significant interactions. Statements of main effects are misleading because they give a picture that all marshes are responding similarly. Where ANOVAs have been conducted, explicit statements regarding statistically significant interactions, or the lack thereof, are necessary to better interpret the study's inferences.

**AR:** Thank you for your comment. We have revised this section discussing the ANOVA results:

"While both marsh type (*p = 1.48e-9*) and marsh zone (*p = 5.62e-20*) yielded significant influence on shear strength values, the interaction between these variables was also significant (*p = 4.67e-11*). This result from the ANOVA indicates that the effect of marsh zone on shear strength varied with marsh type. Interior sites yielded higher values of shear strength than edge sites in the brackish and the salt marshes (salt: 5.1 kPa at edge, 18.5 kPa in interior; brackish: 4.5 kPa at edge, 16.6 kPa in interior) (Fig. 4). There was a negligible difference between edge and interior shear strength values at the freshwater marsh sites (5.41 kPa at edge, 5.38 kPa in interior) (Fig. 4)." (lines 145-150)

**RC2:** 3. Lines 147-148: "The most substantial difference between edge and interior shear strength values occurred at the salt marsh sites, with an increase from 5.1 kPa at the edge to 18.5 kPa in the interior (Fig. 4)."
Reviewer Comment: The difference presented is not likely statistically significant. If this is so, why state it. If fact, it looks to me that the absolute difference between edge and interior for the brackish marsh is of the same magnitude as that for the salt marsh.

**AR:** We have revised this statement to include mention of brackish marshes: "Interior sites yielded higher values of shear strength than edge sites in the brackish and the salt marshes (salt: 5.1 kPa at edge, 18.5 kPa in interior; brackish: 4.5 kPa at edge, 16.6 kPa in interior) (Fig. 4)."

**RC2:** 4. Lines 150-151: "Belowground biomass had the most significant influence on shear strength in the marsh interior ($R^2$ = 0.58, p = 1.09e-5) (Fig. 5)."
Reviewer Comment: Figure 5b shows clearly that this statement is indeed true for salt and brackish marshes, but not for freshwater marshes. This should be clearly stated in the results. The slopes of the three lines in Figure 5b could be statistically compared to confirm that the fresh marshes differ from the salt and brackish marshes, which do not differ.

**AR:** We have clarified this point by adding the phrase "for salt and brackish marshes" after "interior" (line 151).

**RC2:** 5. Lines 151-152: Insert "(data not shown)" after biomass in line 151.

**AR:** We have added "(data not shown)" after line 151 (now line 152).

**RC2:** 6. Lines 188-189: "Nevertheless, our findings indicate that belowground biomass drives soil shear strength variability in the marsh interior (Fig. 5), and soil properties influence marsh edge shear strength (Fig. 6)."
Reviewer Comment: In addition to the soil properties controlling the edge shear strength, isn't low root biomass, especially low root mass per unit volume of soil), found

at virtually all edge sites another important factor (Figure 5a). In fact, it may equally important as soil water content. This point needs mention in the Discussion both here and again in lines 211-212.

**AR:** We have added: "Low concentrations of belowground biomass present at the marsh edge (Fig. 5a) in tandem with processes actively reworking sediment may also contribute to lower soil shear strength values (Silliman et al., 2019)." (line 196-197) in addition to revising lines 211-212 (now lines 227-228): "...we suggest that processes (e.g., sediment deposition, erosion, and resuspension) and environmental conditions (e.g., low belowground biomass, eutrophication, etc.) associated with a more dynamic marsh edge obscure patterns that would otherwise be evident."

**RC2:** 7. Line 231: Delete "along the York River"

**AR:** We have deleted "along the York River" from line 231 (now line 244).

**Additional Revisions:**

- Line 12: added "in" before "biodiverse freshwater marshes"
- Line 105: added "these segments" before "over a 1 mm..."
- Line 106: changed "Live belowground biomass was" to "Samples were then"
- Lines 279-280: added & revised "The authors would like to thank the two anonymous referees for their useful comments that helped improve this manuscript. We would also like to thank the Pamunkey Indian Tribe and the Chesapeake Bay National Estuarine Research Reserve System..." to the acknowledgements.

**References:**

Silliman, B. R., He, Q., Angelini, C., Smith, C. S., Kirwan, M. L., Daleo, P., Renzi, J. J., Butler, J., Osborne, T. Z., Nifong, J. C., and van de Koppel, J.: Field Experiments and Meta-analysis Reveal Wetland Vegetation as a Crucial Element in the Coastal Protection Paradigm, Curr. Biol., 29, 1800-1806.e3, https://doi.org/10.1016/j.cub.2019.05.017, 2019.